# Learning to Evict from Key-Value Cache

**Luca Moschella** [1]   **Laura Manduchi** [1]   **Ozan Sener** [1]

## Abstract

The growing size of Large Language Models (LLMs) makes efficient inference challenging, primarily due to the memory demands of the autoregressive Key-Value (KV) cache. Existing eviction or compression methods reduce cost but rely on heuristics, such as recency or past attention scores, which serve only as indirect proxies for a token's future utility and introduce computational overhead. We reframe KV cache eviction as a reinforcement learning (RL) problem: learning to rank tokens by their predicted usefulness for future decoding. To this end, we introduce KV Policy (KVP), a framework of lightweight per-head RL agents trained on precomputed generation traces using only key and value vectors. Each agent learns a specialized eviction policy guided by a holistic reward, derived from future utility, that evaluates the quality of the ranking across all cache budgets, requiring no modifications to the underlying LLM or additional inference. Evaluated across two model families on the long-context benchmark RULER (up to 128K tokens) and the multi-turn dialogue benchmark OASST2-4k, KVP significantly outperforms strong baselines. Zero-shot tests on standard downstream tasks (BoolQ, LongBench passage retrieval, GovReport) further show that KVP generalizes beyond its training distribution and to considerably longer sequence lengths. These results demonstrate that learning to predict future token utility is a powerful and scalable paradigm for adaptive KV cache management.

## 1. Introduction

Large Language Models (LLMs) based on the Transformer architecture (Vaswani et al., 2017) have revolutionized natural language processing, demonstrating remarkable capabili-

ties across a wide range of tasks (Brown et al., 2020; Önden & Alnour, 2023; Touvron et al., 2023). However, their practical deployment, especially for applications involving long sequences or interactive sessions, is often burdened by substantial computational requirements during inference. A critical bottleneck arises from the Key-Value (KV) cache, a mechanism inherent to autoregressive generation in Transformers (Sheng et al., 2023). This cache stores the keys and values of previous tokens, avoiding redundant computations but growing linearly with the input and generated sequence length. For long contexts, the KV cache can consume tens or even hundreds of gigabytes of memory, rapidly exceeding the capacity of modern hardware accelerators and necessitating strategies for efficient management.

To address this memory challenge, several KV cache management techniques have been proposed. These range from simple heuristics like keeping only the most recent tokens (Xiao et al., 2024), to more sophisticated methods that leverage insights into attention patterns. Approaches like H2O, SnapKV, TOVA, and KeyFormer (Zhang et al., 2023; Li et al., 2024; Oren et al., 2024; Adnan et al., 2024) use signals such as past attention scores or attention sinks to identify and retain important tokens, while attention-free variants leverage key-vector statistics (Devoto et al., 2024; Park et al., 2025; Liang et al., 2025). Others employ quantization to reduce the memory footprint (Sharma et al., 2025; Dettmers et al., 2022; Xiao et al., 2023) or use low-rank approximations to represent the cache state more compactly (Singhania et al., 2024; Ribar et al., 2024). Among these complementary directions, *eviction*, selectively discarding tokens deemed less useful from the cache, is the focus of this work.

While these hand-crafted policies have advanced the state of the art, they are fundamentally built on heuristics that serve as indirect proxies for a token's future importance. They assume that what was important before will remain important. This assumption might not always hold, as the informational needs of generation are dynamic and content-dependent. A token's true value is determined by its utility to future decoding steps, a quantity these methods do not directly optimize for. Consequently, they are prone to suboptimal eviction decisions, leading to an irrecoverable loss of critical information and degraded generation quality. Even post-hoc

---

[1]Apple. Correspondence to: Luca Moschella <luca_moschella@apple.com>.

*Proceedings of the 43rd International Conference on Machine Learning*, Seoul, South Korea. PMLR 306, 2026. Copyright 2026 by the author(s).

Code is available at: github.com/apple/ml-learning-to-evict

| KVP | Future Attention Importance | StreamingLLM |
|-----|------------------------------|--------------|

*Figure 1.* **Future token importance for KV cache eviction.** Effective KV cache eviction requires identifying tokens that will receive little or no future attention. *(center)* We roll out a sample KV cache and measure the true cumulative future attention for each token, then rank tokens by this importance and color them accordingly (bright = high rank, white = low). *(right)* Importance estimated by the fixed sink-and-recency heuristic of *StreamingLLM* deviates substantially from true importance ranking. *(left)* Our learned policy closely recovers the complex, non-local structure of future attention despite using only past keys and values, without access to queries, attention scores, or future tokens. *We expand the comparison with all strategies in Figure 14.*

compression methods based on attention signals are fundamentally backward-looking and have additional drawbacks: they require repeated computation of attention statistics, which introduces significant overhead; and they are not compatible with efficient implementations like FlashAttention (Dao et al., 2022), limiting their practicality.

In this work, we move beyond hand-crafted policies and reframe KV cache eviction as a Reinforcement Learning (RL) problem: learning an attention-free policy that selects which tokens to evict. We formulate this learning problem as a ranking task, where the agent orders tokens by their predicted future utility. As qualitatively illustrated in Figure 1, a learned policy can successfully approximate a token's future utility, a complex, non-local structure, where simple heuristics fail. Such a ranking enables a highly efficient and flexible strategy: eviction is performed by discarding the lowest-ranked entries to meet any memory budget. As a reward signal, we define the ranking successful if for any given cache size, the most valuable information is retained, thus minimizing performance degradation.

We introduce KV Policy (KVP), a framework that trains a distinct, lightweight RL agent for each KV head in the model. This per-head specialization allows each policy to adapt to the unique attention patterns of its corresponding head. The agents are trained efficiently on pre-computed generation traces without any additional inference, using only the key and value vectors as input and requiring no architectural changes to the underlying LLM. To train the agents, we introduce a reward that evaluates the quality of the policy's sorting for all possible cache budgets. For every cache budget, we measure how much relevant information in the future would be erroneously evicted with the candidate sorting when only the top tokens from the cache are kept.

Evaluated across two different model families on long-context synthetic benchmark RULER (Hsieh et al., 2024) and a multi-turn natural language benchmark OASST2-4k (Köpf et al., 2023), KVP significantly outperforms strong heuristic baselines, demonstrating that learning specialized policies to predict the future utility of tokens is a powerful and scalable paradigm for KV cache eviction. Evaluation in a zero-shot generalization setting on downstream benchmarks from the EleutherAI Evaluation Harness (Gao et al., 2024) shows that KVP retains strong performance out of distribution, even to $32\times$ longer context lengths.

In summary, our main contributions are:

- We reframe KV cache eviction as a learning problem: ranking cache entries by their predicted future utility.

- We introduce KVP, a system of lightweight, per-head RL agents that learn specialized sorting policies using only keys and values without any attention information.

- We propose a reward that evaluates eviction policies across all cache budgets without any LLM inference.

- We show that KVP substantially improves long-context performance over strong baselines, scaling to contexts up to 128K tokens, and generalizes to unseen domains.

## 2. Related Work

**Attention-Based Eviction.** Eviction-based approaches aim to choose a subset of the KV cache. Early methods used simple heuristics like First-In-First-Out (FIFO) or Least Recently Used (LRU) (Xiao et al., 2024). More recent work leverages the inherent sparsity in attention patterns. Techniques like StreamingLLM (Xiao et al., 2024) and H2O (Zhang et al., 2023) observe that initial tokens ("attention sinks") and recent tokens often capture most of the required context, allowing for the eviction of intermediate tokens. Others, such as KeyFormer (Adnan et al., 2024) and related works (Cai et al., 2024; Chari & Durme, 2025; Devoto et al., 2025), explicitly analyze attention scores or structures to identify and retain important tokens, sometimes using the current query to inform eviction (Zhang et al., 2025a; Li et al., 2024; Lee et al., 2024a). Although our work is an eviction methodology, it departs from this paradigm by instead learning a *forward-looking, query-independent policy* to directly predict a token's future utility, rather than inferring it from past attention or the current query.

**Memory Hierarchy Management.** Some approaches utilize system memory (CPU RAM) as a secondary cache layer

(Chen et al., 2024b; Sheng et al., 2023) to the GPU memory. Recent sparse retrieval approaches, such as IceCache (Anonymous, 2025), ArkVale (Chen et al., 2024a), MagicPig (Chen et al., 2025), and InfiniGen (Lee et al., 2024b), manage these hierarchies by offloading KV entries to slower memory and retrieving them only when needed, possibly in pages (Tang et al., 2024). While this allows for larger cache sizes, it introduces significant latency (a reload cost) when accessing offloaded entries. Policies for deciding what and when to offload are often heuristic. While our work focuses on eviction, the learned ranking it produces provides a principled, data-driven signal for such hierarchical management: the lowest-ranked entries are natural candidates for being moved to slower memory, representing a powerful synergy between the two approaches.

**Representation Compression.** Instead of removing entries, another line of work focuses on reducing the memory required per entry. Quantization techniques reduce the numerical precision (e.g., to 8-bit integers or lower) of keys and values, cutting memory usage, often with minimal performance impact (Sharma et al., 2025; Dettmers et al., 2022; Xiao et al., 2023), while low-rank approximation methods represent the key and value matrices using lower-dimensional projections (Chang et al., 2025; Singhania et al., 2024; Ribar et al., 2024). State merging approaches like MorphKV (Ghadia et al., 2025) identify and combine similar KV pairs into new, synthetic representations. These techniques are orthogonal and complementary to eviction strategies, as one could utilize our approach to filter out the least useful tokens and then apply MorphKV's merging logic to the remaining high-utility tokens.

**Learned Approaches.** Applying machine learning to directly optimize cache management policies is less common than heuristic approaches. While learning has been used extensively for general caching problems (Afrin et al., 2024; Shuja et al., 2020; Wang & Friderikos, 2020), its application specifically to the dynamic nature of the Transformer KV cache is emerging (Chari et al., 2025; Cetin et al., 2024; Nawrot et al., 2024; Ge et al., 2024). Some recent works, such as Gisting Token (Deng et al., 2025) and Activation Beacon (Zhang et al., 2025b), learn to compress context into compact summary tokens or condensed activations. Our work differs by learning a fine-grained ranking over all tokens rather than a summary; however, these approaches are complementary, as KVP could identify which tokens are best suited for summarization. Our approach is distinguished by utilizing RL to train a lightweight, per-head policy. A novel reward signal, the eviction error across all cache budgets, ensures robust performance under varying memory limits. Furthermore, the per-head rankings and future attention estimates from KVP could directly inform a dynamic, non-uniform budget allocation, a promising direction for future work. Closer to our setting, LOCRET (Huang

et al., 2025) injects retaining heads into each attention layer that take $[Q, K, V]$ as input; since query vectors are not stored in the KV cache, this couples eviction to the LLM's forward pass, whereas KVP operates solely on cached keys, values, and positions and can be applied as a drop-in post-hoc module. The concurrent work JudgeQ (Liu et al., 2026) is methodologically the closest learned baseline: it uses the same future-attention signal as KVP, but trains learnable query embeddings via supervised MSE regression on attention maps inside the LLM, still requiring full forward passes, while KVP trains fully offline on pre-collected traces. Similar in philosophy, the concurrent work (Jegou & Jeblick, 2026) approximates KV importance scores estimated by (Kim et al., 2025), that requires two LLM forward passes.

Overall, while prior work has addressed KV cache constraints through heuristics, memory hierarchies, or compression, our approach introduces an RL framework that casts eviction as a ranking problem. By training lightweight, per-head policies to predict tokens' future utility and optimizing them with a global, budget-agnostic reward, we offer an adaptive, offline, query-independent solution. Crucially, this method is also complementary to many existing techniques, and may provide a principled signal for memory offloading, context gisting, or dynamic budget allocation.

## 3. KV Policy (KVP)

We consider the problem of KV cache eviction: given $n$ tokens $\mathcal{X} = \{x_i\}_{i \in [n]}$ and a budget $b$, select a subset $\mathcal{S}^\star \subset \mathcal{X}$ maximizing some downstream performance reward $R$ as

$$\mathcal{S}_b^\star \in \arg\max_{|\mathcal{S}|=b, \mathcal{S} \subset \mathcal{X}} R(\mathcal{S}). \tag{1}$$

Because generation is autoregressive and budgets vary over time, the solution must handle arbitrary $n$ and all $b \in [n]$. For a general reward $R$, this is a hard combinatorial selection problem. Rather than tackle it in full generality, we *design* our reward so that the optimum is tractable by construction, satisfying **(i) uniqueness:** each $\mathcal{S}_b^\star$ is unique, ensured via simple tie-breaking; and **(ii) nestedness:** $\mathcal{S}_b^\star \subset \mathcal{S}_{b+1}^\star$ for all $b$. Nestedness does not hold for arbitrary task rewards, but holds in our setting because the reward (Section 3.1.1) is additive over tokens. Under these conditions, KV cache eviction is equivalent to KV cache ranking *with proof deferred to Appendix A.1.*

**Proposition 3.1.** *Assume (i) **uniqueness** of each $S_b^\star$ and (ii) **nestedness**: $S_b^\star \subset S_{b+1}^\star$ for all $b$. Then, there exists a total order (ranking) $\pi$ such that*

$$S_b^\star = \{\, i \in [n] : \pi(i) \leq b \,\} \quad \text{for all } b.$$

*Equivalently, there exists a scoring function whose top-$b$ elements realize $S_b^\star$ for every $b$.*

Following proposition 3.1, we reformulate eviction as learning a single budget-agnostic scoring function. For a given

budget $b$, we sort the cache entries from most to least valuable for future decoding using the learned scoring function and retain the top-$b$. A high-quality ranking preserves critical information across all budgets, minimizing degradation.

We use RL to learn this scoring function. We parameterize a stochastic ranking policy and directly optimize the true discrete end-to-end reward using policy-gradient methods. We analyze this choice against differentiable relaxations of sorting (Prillo & Eisenschlos, 2020; Grover et al., 2019; Blondel et al., 2020) in Section 4.2.

We employ a lightweight RL agent, governed by parameters $\theta$, to define the scoring function $f(;\theta)$. To capture the specialized functions of different attention mechanisms, we train a distinct agent for each KV head in the LLM. To this end, we introduce KVP, a framework for efficiently training lightweight, per-head RL agents to perform this ranking. In the remaining of this section, we first discuss the agent architecture, the reward, and finally the learning process.

### 3.1. KV Cache Eviction Agent

We follow the Plackett-Luce model (Plackett, 1975) to formulate learning to sort as an RL problem. Given a set of tokens $\{x_i\}_{i \in [N]}$, we learn a parametric scoring function $f(x_i; \theta)$ which assigns a score to each token $x_i$. This scoring function induces a stochastic sorting policy $\pi_\theta$ which samples permutation $\sigma = (\sigma_1, \ldots, \sigma_N)$ sequentially as:

$$\pi_\theta(\sigma|x_1, \ldots, x_N) = \prod_{i=1}^{N} \frac{\exp\left(f(x_{\sigma_i}; \theta)\right)}{\sum_{j=i}^{N} \exp\left(f(x_{\sigma_j}; \theta)\right)} \quad (2)$$

At each step $i$, the next element $\sigma_i$ is sampled proportionally to its score, normalized over the remaining tokens. This process defines a valid distribution over all permutations. The scoring function $f(;\theta)$ together with the sampling policy, forms our KV cache eviction agent. We next specify the parameterization of $f(;\theta)$.

**Scoring Function** The representation for token $x_i$ is the concatenation of its key vector $k_i$, value vector $v_i$, and its original position $pos_i$ as $x_i = (k_i, v_i, pos_i)$. We define the scoring function $f(;\theta)$ as a small MLP parametrized with $\theta$ as $f(x_i \theta) = MLP_\theta(k_i, v_i, pos_i)$. Crucially, the policy relies only on information available in the cache and does not require access to future information, previous attention scores or any query embedding.

**Efficient Parallel Sampling with Gumbel-Sort** While Equation (2) defines the distribution, sequential sampling is inefficient. Fortunately, a permutation can be sampled from this distribution in a single step using the Gumbel-Sort (Mena et al., 2018). Given the scores $f(x_i; \theta)$, we generate i.i.d. noise samples $g_i \sim \text{Gumbel}(0, 1)$. A permutation $\sigma$ is then sampled by sorting the perturbed scores:

$$\sigma = \text{argsort}_{i \in [N]} \left(f(x_i; \theta) + g_i\right) \quad (3)$$

This procedure is non-autoregressive, fully parallelizable on modern hardware, and allows us to sample an entire permutation with just one forward pass of the scoring model and a fast sort operation. This is critical for efficient training.

#### 3.1.1. GLOBAL REWARD FOR OFFLINE RL

A key component of our framework is a reward signal that globally evaluates the quality of an agent's entire ranked output, directly optimizing for the efficient preservation of information across all possible cache budgets. More importantly we define this reward in a way to enable training without any additional LLM inference.

Consider the input set of $n$ tokens $\mathcal{X} = \{x_i\}_{i \in [n]}$ and $f$ future tokens $x_{n+1}, \ldots, x_{n+f}$ in the training data, and let $A(x_i, x_j)$ denote the attention from $x_j$ to $x_i$. For models with Grouped-Query Attention (Ainslie et al., 2023), the attention score $A(x_i, x_j)$ is the maximum attention value across all queries within that group. Given the permutation $\sigma$, a cache of budget $b$ retains the top-$b$ tokens $\sigma_1, \ldots, \sigma_b$ and evicts the rest. The per-budget cost is the total future-attention importance of the evicted tokens,

$$\mathcal{C}^b(\sigma_{b+1}, \ldots, \sigma_n; \mathcal{X}) = \sum_{i=b+1}^{n} \sum_{j=n+1}^{n+f} A(x_{\sigma_i}, x_j), \quad (4)$$

and the total cost of a ranking is the area under this cost-vs-budget curve, which equivalently rewrites as a rank-weighted sum:

$$\mathcal{C}(\sigma; \mathcal{X}) = \sum_{b=0}^{n-1} \mathcal{C}^b = \sum_{b=1}^{n} b \cdot \sum_{j=n+1}^{n+f} A(x_{\sigma_b}, x_j), \quad (5)$$

since $\sigma_b$ is evicted at all $b$ budgets smaller than its rank. To make training scale-invariant, we normalize by the cost of the optimal ranking $\sigma^\star = \arg\min_\sigma \mathcal{C}(\sigma; \mathcal{X})$, and define the per-budget reward

$$\mathcal{R}^b(\sigma_b; \mathcal{X}) = -\frac{b \cdot \sum_{j=n+1}^{n+f} A(x_{\sigma_b}, x_j)}{\mathcal{C}(\sigma^\star; \mathcal{X})}. \quad (6)$$

The total reward equals the negated normalized AUC,

$$\mathcal{R}(\sigma; \mathcal{X}) = \sum_{b=1}^{n} \mathcal{R}^b = -\frac{\mathcal{C}(\sigma; \mathcal{X})}{\mathcal{C}(\sigma^\star; \mathcal{X})} \leq -1, \quad (7)$$

with equality at $\sigma^\star$. Maximizing $\mathcal{R}$ minimizes the normalized AUC. Figures plot the per-budget cost $-\mathcal{R}^b \geq 0$, the contribution of $\sigma_b$ to the normalized AUC (lower is better).

---

**Algorithm 1** RL Training on Pre-Computed KV Traces

---

1: **Input:** Static dataset of pre-computed traces over $m$ sequences each with length $n_j, j \in [m]$ denoted as $\mathcal{X}^j = \{x_i^j\}_{i \in n_j}$ containing Q, K, V tensors.
2: **while** training not converged **do**
3:    Sample a data item $j \sim \text{Unif}(m)$ and a context length $n \sim \text{Unif}(n_j)$.
4:    Sample $K$ permutations from the agent

$$\sigma_1^k, \ldots, \sigma_n^k \sim \pi_\theta(\sigma | x_1^j, \ldots, x_n^j), k \in [K]$$

5:    Calculate reward $\mathcal{R}(\sigma_1^k, \ldots, \sigma_n^k; \mathcal{X}^j)$ using (7).
6:    Update $\theta$ using (8).
7: **end while**

---

### 3.2. Per-Head RL Agent and Efficient Training

A significant advantage of our method is its training efficiency. The agents are trained entirely offline, obviating the need for live LLM inference within the RL optimization loop. This is possible because our reward is computed using the static future tokens, rather than dynamically generated text. First, we generate a dataset by running the base LLM on a training corpus. For each sample, we store the full sequence of queries, keys, and values. The attention matrices are not stored due to their prohibitive size.

During training, we sample a sequence and a cache size to be evicted from. We update the parameters $\theta$ of the agent using a policy gradient algorithm. Since our reward $\mathcal{R}(\sigma)$ is a terminal reward assigned only after the entire permutation $\sigma$ is generated, the objective is $J(\theta) = \mathbb{E}_{\sigma \sim \pi_\theta}[\mathcal{R}(\sigma)]$. We use the REINFORCE algorithm with a Leave-One-Out (RLOO) baseline (Williams, 1992; Ahmadian et al., 2024) to reduce variance. For an episode (permutation) $\sigma^k$ within a batch of $K$ episodes, the baseline $\bar{\mathcal{R}}$ is the average terminal reward of all other episodes in the batch. Considering the advantage, $\mathcal{R}(\sigma^k) - \bar{\mathcal{R}}$, the gradient is thus estimated as:

$$\nabla_\theta J(\theta) \approx \frac{1}{K} \sum_{k=1}^K \left[ (\mathcal{R}(\sigma^k) - \bar{\mathcal{R}}) \sum_{i=1}^n \nabla_\theta \log \pi_\theta(\sigma_i^k | \mathcal{X}) \right]. \tag{8}$$

We summarize the training in Alg. 1 and defer the further implementation details to Section A.2. Training per-head agents from offline traces is highly scalable. At inference, the learned policy for each head ranks its KV entries, and the trailing entries are evicted based on the budget.

## 4. Evaluation

We conduct a comprehensive evaluation of KVP across multiple language modeling benchmarks and a wide range of cache budgets. Our results show that KVP consistently shows stronger downstream performance relative to existing heuristics, including methods that exploit privileged, query-specific information. This results confirm that high-quality KV cache eviction policy can be trained entirely offline and deployed at inference time using only static token features (keys, values, and positions). We also provide ablation studies to analyze the design choices behind KVP.

**Base Model and Datasets.** Our experimental setup is centered on `Qwen2.5-7B-Chat` (Yang et al., 2024; Wang et al., 2024). We include further evaluation using `Phi-4 14B` (Abdin et al., 2025) in the Appendix, Figure 13. We train and evaluate our KVP agents using two distinct long-context benchmarks: *RULER-4k* (Hsieh et al., 2024), a synthetic dataset designed to probe long-context reasoning with sequences of approximately 4500 tokens; and *OASST2-4k*, a subset of the OpenAssistant Conversations Dataset (Köpf et al., 2023) featuring multi-turn dialogues of similar length.

To assess generalization, we evaluate on extended context lengths using *RULER-128K*, as well as the *GovReport* summarization and *Passage Retrieval* (English and Chinese) tasks from the LongBench benchmark (Bai et al., 2023). We further perform zero-shot evaluation on four standard downstream tasks from the EleutherAI Language Model Evaluation Harness (Gao et al., 2024): *BoolQ* (Clark et al., 2019), a reading comprehension task; *ARC-Challenge* (Clark et al., 2018), requiring multi-step science reasoning; *MMLU* (Hendrycks et al., 2021), testing expert-level knowledge; and *HellaSwag* (Zellers et al., 2019), a commonsense reasoning task. Importantly, KVP is not trained or fine-tuned on any of these generalization tasks. We defer further implementation details to Section A.2.

**Baselines.** We benchmark KVP against both attention-based and attention-free KV cache management techniques. The first category includes state-of-the-art methods like *TOVA* (Oren et al., 2024) and *SnapKV* (Li et al., 2024). These methods leverage attention scores to identify important tokens, which introduces computational overhead by tying the eviction strategy to the attention calculation step. The second, more efficient category of attention-free baselines, to which our own KVP agents belong, operates independently of the attention mechanism. This group includes a *Random* eviction baseline, the recency-based *StreamingLLM* (Xiao et al., 2024), and several approaches that prune tokens based on their key embeddings. These include methods based on statistical patterns (*LagKV* (Liang et al., 2025)), vector similarity (*KeyDiff* (Park et al., 2025)), and L2 norm (*K-Norm* (Devoto et al., 2024)). Additionally, we report a comparison against the concurrent learned baseline *JudgeQ* (Liu et al., 2026) in Section A.4, where KVP consistently outperforms JudgeQ across RULER-128K, LongBench passage retrieval (EN/ZH), and BoolQ. To ensure a fair comparison, we adapt all baselines to our ranking-based framework by

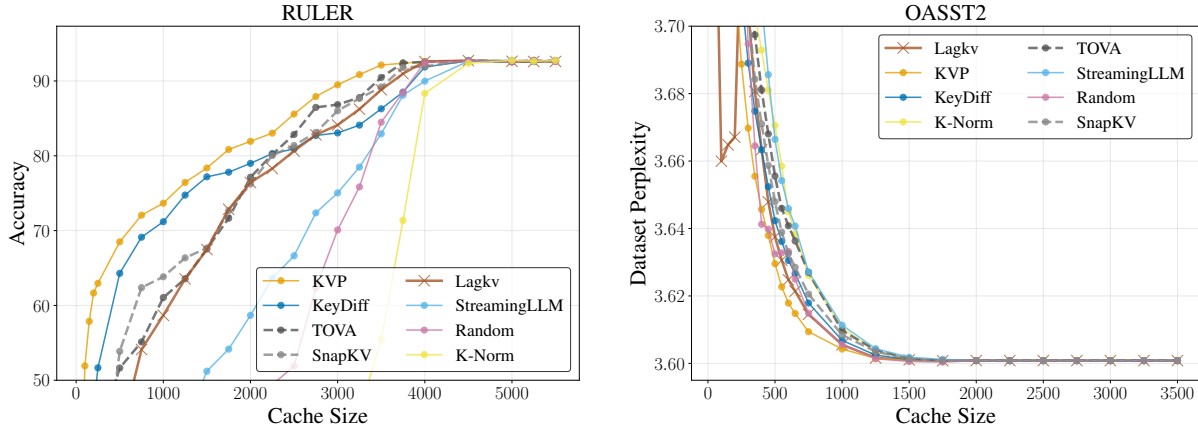

*Figure 2.* Overall in-distribution accuracy on the RULER-4k benchmark (Left) and perplexity on the OASST2-4k test set (Right), as a function of the absolute KV cache size. KVP achieves the highest accuracy and lowest perplexity across most of the possible cache sizes.

converting their binary keep/evict decisions into a full token permutation, enabling consistent evaluation.

**Budgeting and Compression Schedule.** For all online evaluations, we follow one of two consistent compression protocols. For short-context benchmarks (RULER-4K, OASST2-4k, BoolQ, ARC-Challenge, GovReport), we use a *single compress-after-prefill* schedule: the context is processed by the LLM to populate the initial KV cache, the specified compression method is then applied once to reduce the cache to the target budget, and the model finally generates the response autoregressively using the compressed cache. For long-context benchmarks (RULER-128K, Long-Bench passage retrieval), a single-shot prefill would exceed the available memory, so we instead adopt a *chunked prefill-compress* loop: the input is processed in fixed-size chunks, and after each chunk the compression method is re-applied to the *union of the previously kept cache and the newly pre-filled tokens*, keeping the total cache size at the target budget throughout. Generation then proceeds from the compressed cache as in the short-context protocol. Both protocols therefore evaluate the same underlying eviction policy; the chunked variant additionally stresses repeated application of that policy across the prefill. To isolate the performance of the core ranking strategy, we apply a uniform compression budget across all heads and layers for all methods. This ensures a fair comparison focused purely on the quality of the eviction strategy itself, rather than on budget allocation heuristics. While head-specific or layer-specific budgeting is an orthogonal and promising direction for further performance improvements, our approach provides a clear and interpretable evaluation of the underlying policies.

**Inference Efficiency.** Our KVP agents make their ranking decisions using only the Key and Value vectors, and their position in the context. This makes KVP highly efficient, as it avoids re-computing attention scores. In contrast, attention-based baselines are evaluated using the attention

scores generated during the prefill stage, giving them access to information about how the final user message, for example, attends to the rest of the context. Our method is therefore benchmarked against baselines that have access to more direct, query-specific information at the time of compression. We highlight in all the figures the attention-based baselines with dashed lines. Please refer to Section A.3 for FLOPs estimation and wall-clock benchmarks.

**Absolute vs. Relative Cache Size.** A final methodological note concerns our use of cache size. We report performance as a function of absolute KV cache size (i.e., the number of tokens retained) rather than a relative compression ratio. This decision is motivated by practical application: practitioners operate under fixed memory constraints, making an absolute token budget a more direct and interpretable measure of resource cost. In contrast, a compression ratio's impact on memory is dependent on the initial context length, making it a less stable metric for cross-scenario comparison.

### 4.1. Downstream Performance

We apply the strategies to a live LLM, reducing the KV cache to a target budget after the prefill before measuring performance on several downstream benchmarks.

**RULER benchmark.** We evaluate performance on the RULER-4k benchmark using its official text-based accuracy metric, which requires generating the correct answer for long-context reasoning tasks. This tests whether the methods preserve the specific, often non-local, information needed for complex problem-solving. Figure 2 (left) shows that KVP consistently outperforms all baselines, retaining higher accuracy as the cache budget shrinks. This result is a direct consequence of its learned policy: unlike heuristics that might discard old but crucial clues, KVP learns to identify and keep these high-value tokens, regardless of their position, enabling the model to succeed at the task. A

detailed breakdown of performance across all RULER sub-tasks, along with their corresponding eviction error curves, is provided in Figure 20, further highlighting the robustness of our method. This result is particularly noteworthy given that RULER's structure heavily favors strategies that can use the final question to identify relevant context; an advantage held by attention-based baselines but not by KVP.

**OASST2 benchmark.** We evaluate the efficacy of KV cache compression by its impact on perplexity (PPL), a measure of the model's next-token prediction capability. An effective compression method should minimize PPL degradation. As shown in Figure 2 (right), KVP achieves lower perplexity than baselines that rely solely on KV embeddings, except for *LagKV* at the very smallest budgets. Furthermore, it often outperforms methods that use additional information across nearly all cache sizes. In contrast, heuristic approaches like *StreamingLLM* exhibit a sharp increase in PPL as the cache budget decreases, confirming the brittleness of their fixed-rule strategies. The superior performance of KVP demonstrates a direct link between our training objective and the preservation of the model's downstream capabilities. These findings are further corroborated by PPL results on the RULER benchmark (Appendix Figure 16), where KVP maintains a similar advantage.

**Generalization to longer contexts.** We stress-test the policy by evaluating at a context length $32\times$ longer than the one seen during training: on *RULER-128K* (Figure 3), KVP trained at $\sim$4K preserves its lead over every heuristic and attention-based baseline at every cache budget. This length extrapolation indicates that the learned ranking captures intrinsic, length-agnostic properties of token utility rather than artifacts of the training context.

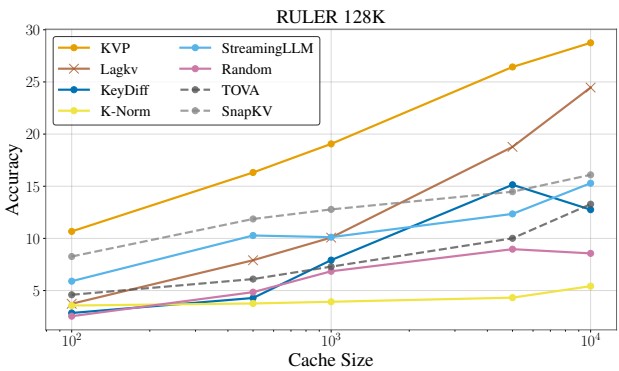

*Figure 3.* Mean accuracy on *RULER-128K* across the 13 subtasks, as a function of the absolute KV cache size. Trained on sequences of $\sim$4K tokens, KVP extrapolates to a $32\times$ longer context and still dominates every baseline at every budget. Evaluated under chunked prefill-compress with physical tensor compaction.

**Zero-shot generalization.** LLMs are widely regarded as general-purpose computational systems and their utility depends on robust performance in out-of-domain scenarios.

Even though KVP enables efficient training of domain-specific policies using only unlabeled sequences, we evaluate its zero-shot generalization performance on a set of standard downstream tasks. Specifically, we test agents trained on RULER-4k ($KVP^R$) and on *OASST2* ($KVP^S$) on *BoolQ*, on the *GovReport* summarization task and the *passage retrieval* task of the LongBench benchmark, reporting performance at varying KV cache sizes. Importantly, the "prefill" stage does not include the question in *BoolQ* and *GovReport*, so as to better reflect real-world scenarios where a given text is compressed for different questions not known in advance. Results on *ARC-Challenge*, *MMLU*, and *HellaSwag* are presented in Section A.7. This evaluation is designed to assess whether policies optimized for long-context efficiency maintain performance on short and long-context benchmarks that probe factual knowledge, multi-step reasoning, and commonsense inference.

The results, presented in Figures 4 and 5, demonstrate that KVP consistently achieves competitive performance. Across the diverse benchmarks, both $KVP^R$ and $KVP^S$ rank at or near the top, outperforming heuristic baselines, with gentle degradation as the cache budget shrinks. A clean specialization pattern emerges, aligned with each agent's training distribution: $KVP^S$ (trained on conversational OASST2 dialogues) holds the advantage on *BoolQ* (a reading-comprehension yes/no QA task) and on the *GovReport* summarization task, while $KVP^R$ (trained on RULER) holds the advantage on *LongBench passage retrieval* in both English and Chinese, a transfer made non-trivial by the fact that LongBench is a separate benchmark sourced from real long-context documents, yet $KVP^R$'s policy carries over because the relevant token-utility structure transfers. This specialization is a feature, not a limitation: it shows that practitioners can train KVP on representative data for their target workload and obtain a policy tailored to it, while still inheriting the broad robustness visible across all benchmarks. This success preserves the model's core capabilities, confirming that KVP is not a specialized tool with significant trade-offs, but a robust technique that can be enabled by default to provide memory savings while minimizing degradation to the model's general utility.

### 4.2. Ablations

To validate the core design choices of KVP, we conduct two key ablation studies. First, we analyze our proposed reward function, as defined in Equation (7). Second, we study our use of RL instead of supervised surrogates of sorting.

**Is the reward function $\mathcal{R}$ effective?** To confirm the reward $\mathcal{R}$ enables effective learning of eviction policies, we measure the negative per-budget reward, $-\mathcal{R}^b$, on unseen test data. This value is the contribution of $\sigma_b$ to the normalized AUC $\mathcal{C}(\sigma)/\mathcal{C}(\sigma^\star)$, which our agent is trained

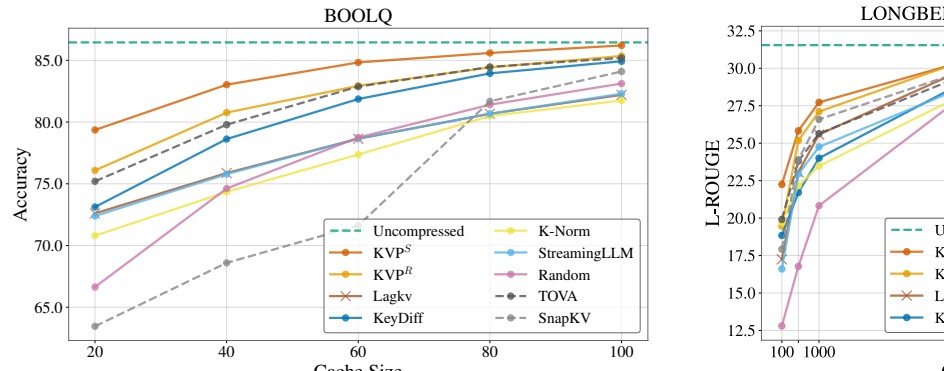

*Figure 4.* Left: average test accuracy on *BoolQ* (higher is better). Right: L-Rouge score on the *GovReport* summarization task from LongBench (higher is better). Both reported as a function of the absolute KV cache size, with both $KVP^R$ and $KVP^S$ shown; $KVP^S$ (OASST2-trained) holds a consistent advantage on both, reflecting its conversational training distribution.

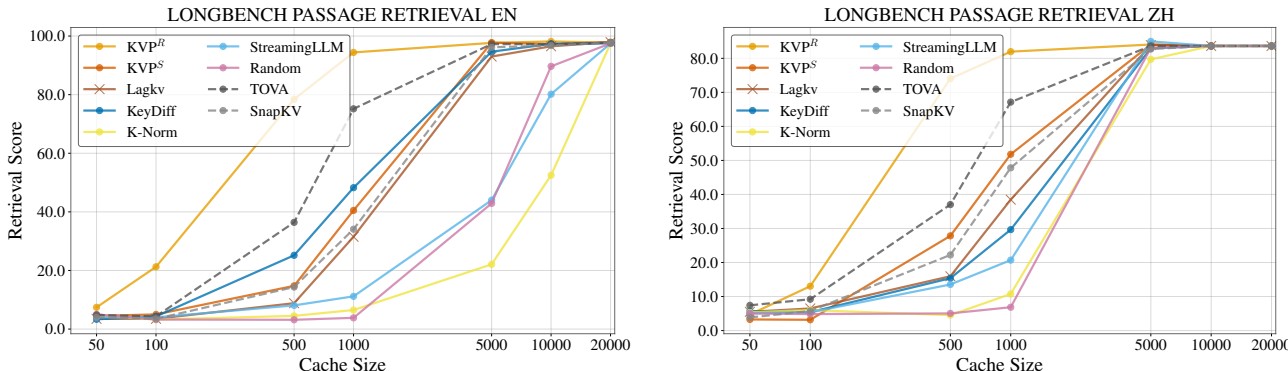

*Figure 5.* Passage retrieval score on the *LongBench* benchmark, English (Left) and Chinese (Right) splits, as a function of the absolute KV cache size. Both $KVP^R$ and $KVP^S$ are shown; $KVP^R$ (RULER-trained) holds the advantage on the retrieval task, consistent with the retrieval-style structure of its training distribution. Evaluated under the chunked prefill-compress protocol with physical tensor compaction.

to minimize across all budgets simultaneously. The results in Figure 6a show a clear separation between methods: while attention-aware methods that use query information at inference time (dashed lines) form a distinct, low-loss cluster, heuristics without attention scores form a high-loss cluster. This is rather expected as the target, future attention scores, are highly related to past attention. Crucially, KVP performs within this top-tier group, achieving a loss comparable to methods like *TOVA* and *SnapKV* without using their privileged information. This demonstrates that our offline RL training successfully distills the principles of attention-based ranking into an efficient, static policy. Furthermore, the analysis reveals the importance of policy specialization across heads. A fixed heuristic like *StreamingLLM* can be effective for certain heads (see layer 22, head 0 in Figure 22) but detrimental for others (see layer 19, head 0 in Figure 21). Even *KeyDiff*, a strong attention-free heuristic, exhibits hard failure modes (see layer 2, head 0 in Figure 22). This result, together with a comprehensive sweeps across all layers and heads in Figures 21 and 22, confirms that KVP learns diverse patterns that cannot be captured with a single heuristic.

**Is RL necessary?** Since our global reward reduces to a rank-weighted sum of per-token importances (Equation (7)), a natural question is whether a supervised learning-to-rank objective could replace RL. We therefore compare KVP against a suite of supervised baselines trained with the same network, data, and input features: (i) *Pointwise Regression* on the per-token importance $u_i = \sum_{j=n+1}^{n+f} A(x_i, x_j)$; (ii) *Pairwise RankNet* (Burges et al., 2005); (iii) *Listwise ListNet* (Cao et al., 2007); (iv) a *Soft-Sort* surrogate that minimizes a length-normalized MSE between predicted and ground-truth ranks through a differentiable sorter (Blondel et al., 2020); and (v) a *Weighted Soft-Sort* variant that rescales per-token rank errors by the ground-truth importance. We use sane defaults and normalizations for all five.

Figure 6b shows a clean tradeoff driven by the heavy-tailed (Zipfian) distribution of attention. *Magnitude-sensitive* losses (Regression, Pairwise, Listwise, and Weighted Soft-Sort) devote capacity to the few attention sinks that dominate the loss: they are competitive at the smallest budgets, where retaining sinks is sufficient, but their cost

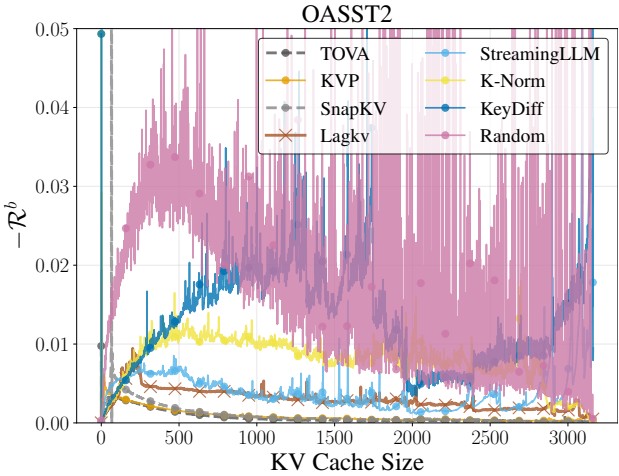

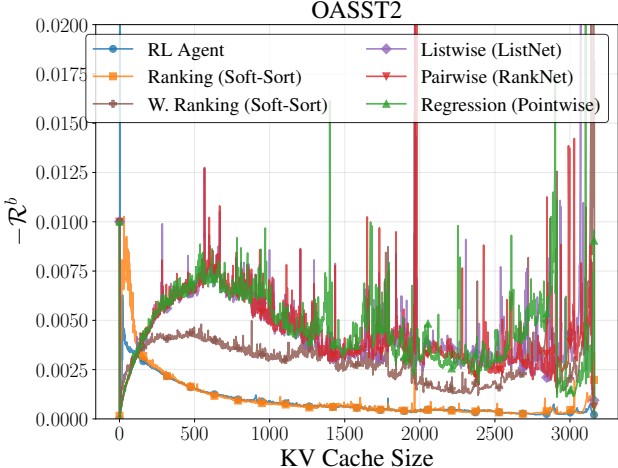

*(a)* **Effectiveness of the reward:** The methods split into two clusters: attention-aware policies that use query information at inference (dashed lines) form a low-cost group, while attention-free heuristics cluster at higher cost. KVP joins the top-tier cluster, matching *TOVA* and *SnapKV* despite not using their privileged information, by learning to predict future attention. A full visualization for all the layers is in Figures 21 and 22.

*(b)* **RL vs. Supervised Learning-to-Rank:** KVP (RL) attains the lowest cost across nearly the entire budget range. *Magnitude-sensitive* losses devote capacity to the few attention sinks that dominate the loss; the *magnitude-invariant* Soft-Sort surrogate matches RL in the bulk of the curve, yet exhibits a pronounced spike at the left, because a rank-only loss penalizes every swap equally even if the top tokens dominate the cost at tight budgets.

*Figure 6.* Cost per-budget ($-\mathcal{R}^b$) on the OASST2 test set for a representative head (layer 10, head 0).

stays well above RL throughout the rest of the curve because they have not learned the ordering of the non-sink tokens. The *magnitude-invariant* Soft-Sort surrogate exhibits the opposite failure mode: it tracks RL closely across the bulk of the curve but spikes sharply at tight budgets (the left of Figure 6b)—exactly where the most important tokens live—because a rank-only loss penalizes every swap equally and therefore cannot distinguish among the top-ranked tokens whose ordering drives the cost when the budget is tight. Weighted Soft-Sort, which might appear to combine the strengths of both, collapses back to the magnitude-sensitive regime: re-introducing magnitude as a per-token weight re-amplifies the sinks.

A rank-dependent rescaling that bridges the two failure modes almost certainly exists in principle, but its shape depends on the per-head and per-input importance distribution, which varies widely across heads (Section A.5) and would need to be re-tuned for every head and cache. Optimizing the eviction-cost AUC directly via policy gradients sidesteps this entirely: each swap is penalized in proportion to its contribution to the budget AUC, so the learning signal is cost-weighted and distribution-agnostic by construction.

## 5. Conclusions

We introduced KVP, framing KV cache eviction as a learnable sorting task. By training lightweight, per-head policies to rank entries by future utility via an inference-free RL objective, we enable adaptive memory management without architectural changes. Our evaluations span short and long-

context benchmarks, including RULER at 128K tokens and LongBench passage retrieval, and show that KVP consistently outperforms heuristic baselines while preserving the underlying model's capabilities, validating that learned sorting policies are more robust than handcrafted importance proxies. More broadly, our findings indicate that effective KV cache compression does not need to rely on either privileged access to inference-time attention or carefully hand-tuned heuristics, but can instead be realized by a compact, head-specific policy learned once and offline.

**Limitations and future work.** KVP makes several simplifying design choices that open avenues for future work. We apply a uniform compression budget across heads and layers, so adaptive, non-uniform allocation is a natural next step. Because we train a separate agent per head, cross-head dependencies are not modeled; joint end-to-end optimization of all policies could capture them, at the cost of requiring online LLM inference during training. Our reward similarly avoids inference, using future attention as a proxy for task-level utility rather than a direct downstream signal; incorporating task feedback into training is a promising extension. Furthermore, KVP's scoring depends only on static token features (keys, values, positions), so scores of already-cached tokens remain valid as the sequence grows and only newly generated tokens need scoring; the approach is therefore architecturally compatible with continuous decoding-time compression. Finally, our current implementation lacks an optimized end-to-end inference pipeline, and we therefore report policy-level wall-clock costs rather than full-system latency.

## Impact Statement

This paper presents work whose goal is to advance the field of machine learning. There are many potential societal consequences of our work, none of which we feel must be specifically highlighted here.

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

# A. Appendix

## A.1. Missing Proof of Proposition 1

*Proof.* Since $|S_{b+1}^\star| = b+1$ and $|S_b^\star| = b$ with $S_b^\star \subset S_{b+1}^\star$, the set difference $S_{b+1}^\star \setminus S_b^\star$ is nonempty. By uniqueness of $S_{b+1}^\star$, this difference must contain exactly one element; otherwise, there would exist two distinct $(b+1)$-subsets strictly between $S_b^\star$ and $S_{b+1}^\star$, contradicting uniqueness. Hence we can define a sequence of distinct elements

$$x_{\sigma_1} \in S_1^\star, \qquad x_{\sigma_{b+1}} \in S_{b+1}^\star \setminus S_b^\star \quad (b = 1, \ldots, n-1).$$

By construction, for each $b$ we have

$$S_b^\star = \{x_{\sigma_1}, \ldots, x_{\sigma_b}\}.$$

Now define a total order (ranking) $\pi$ on $[n]$ by setting

$$\pi(x_{\sigma_b}) = b \quad (b = 1, \ldots, n).$$

This is a bijection $\pi : [n] \to [n]$, and its top-$b$ prefix is precisely $\{x_{\sigma_1}, \ldots, x_{\sigma_b}\} = S_b^\star$. Therefore,

$$S_b^\star = \{\, i \in [n] : \pi(i) \le b \,\} \quad \text{for all } b,$$

as claimed.

Equivalently, given $\pi$ we may define a scoring function consistent with the order, for instance

$$s(x_{\sigma_k}) = n - k \quad (k = 1, \ldots, n),$$

which is strictly decreasing in $k$. Then the top-$b$ elements according to $s$ are exactly $\{x_{\sigma_1}, \ldots, x_{\sigma_b}\} = S_b^\star$. $\qquad\square$

## A.2. Implementation details

Our KVP agents are lightweight 2-layer MLPs with 256 hidden units, trained using the RLOO algorithm (Ahmadian et al., 2024) as described in Section 3. Following the Grouped-Query Attention (GQA) architecture of `Qwen2.5-7B-Chat`, we train a separate agent for each of the 4 KV heads across all 28 layers, yielding 112 specialized agents. Each agent contains approximately 650K parameters.

The agents are optimized to maximize the reward signal in Equation (7), which encourages retention of tokens with high future utility across all cache budgets. For training stability, we apply gradient clipping (maximum norm of 5) and normalize advantages by their mean and standard deviation, without entropy regularization. We use AdamW with a learning rate of $5 \times 10^{-5}$, following a cosine schedule with 100-step linear warmup (start factor 0.01) that decays to $1 \times 10^{-6}$. Each agent trains for 4,000 steps on pre-computed activations.

During inference for evaluation purposes, the learned policy ranks all tokens except the first 4 and last 16, which are always retained. We validate the approach by emulating eviction with custom attention masks in FlexAttention on benchmarks that fit in memory, and use a physical tensor compaction of the retained keys and values for the long-context benchmarks (RULER-128K and LongBench passage retrieval).

## A.3. Training Efficiency and Inference Overhead

The training process for our KVP agents is designed to be highly efficient, imposing minimal computational and storage overhead. Agents are trained on a small dataset of pre-computed activation traces; for our experiments, we used approximately 6,000 samples from the RULER-4K dataset and 4,500 from the OASST2 dataset. Data generation represents the primary one-time cost of this pipeline, requiring a single forward pass over these training samples to collect and store on disk the Query, Key, and Value embeddings for the entire sequence for each KV cache. This step incurs a computational cost approximately equivalent to standard inference on the dataset. For example, trace collection on the OASST2 training split (4,649 samples, 28 layers $\times$ 4 KV heads) completes in under two hours on 7 nodes with 8 GPUs each; the workload is embarrassingly parallel, so each node processes data independently and the full dataset never resides on a single machine. The resulting traces occupy $\sim$1.2 TB on cluster storage, partitioned so that each per-(layer, head) agent only ever loads its own $\sim$11 GB slice.

Each per-head agent is a small MLP with approximately 650K parameters, resulting in a checkpoint size of only 2.6MB. The agent training completes in less than 30 minutes on a single node of 8 NVIDIA H100 GPUs. We note that this training time is achieved without extensive hyperparameter tuning or code optimizations for speed, suggesting that further significant speed-ups are possible. These factors highlight the low computational footprint of our offline training framework.

**FLOPs Estimation**. To assess computational cost independent of hardware optimization (e.g., kernel fusion), we quantify the FLOPs overhead per token:

- Autoregressive Generation (Throughput). Zero Overhead. In the prefill-then-compress setting, the KV cache is compressed only once after prefill. This means KVP introduces no additional FLOPs during the decoding phase. Consequently, its throughput is identical to any other eviction method at the same cache budget. KVP's contribution is achieving superior accuracy for that given level of throughput.

- Prefill Latency. Cost: $14.00$ GFLOPs $\rightarrow 14.15$ GFLOPs. The overhead is strictly confined to the prefill stage. Based on the standard approximation of $2 \times N_{params}$ FLOPs per token:

  - Base model (Qwen-7B): $14.00$ GFLOPs/token.
  - KVP Overhead: Our 112 agents ($0.65$M params each) add $72.8$M parameters, contributing an additional $0.15$ GFLOPs/token.
  - Total: The prefill cost becomes $14.15$ GFLOPs/token, a marginal increase of $1\%$.

**Wall-clock Benchmarks**. In Figure 7, we perform empirical measurements validating these theoretical estimates. The results confirm that KVP compression adds negligible latency, orders of magnitude lower than the LLM prefill time. Furthermore, as noted in the figure, the independence of KVP operations per cache allows for easy parallelization, ensuring the practical overhead remains minimal.

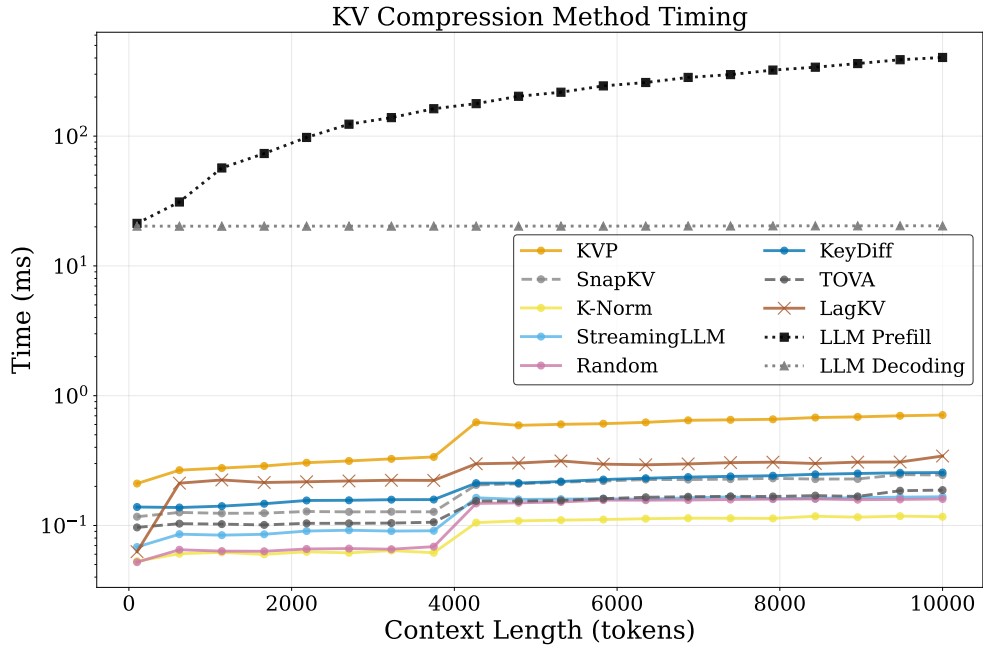

*Figure 7.* Wall-clock latency of eviction decisions (logarithmic y-scale). We isolate the computational overhead of each strategy by measuring only the time required to compute importance scores and determine eviction indices for a single KV head. We exclude the compressed LLM inference time because, once eviction indices are determined, LLM throughput depends only on the cache size, not the selection policy. We compare compression times against Qwen2.5-7B-Instruct baselines: LLM Prefill (time to process context) and LLM Decoding (time to predict a single subsequent token). KVP adds negligible overhead: at 10k context length, a single-layer compression takes 0.71ms vs. 404ms for the full-model prefill (a 570× difference). Since KVP operates independently on each KV cache and relies only on keys/values (not the current query), it allows for easy parallelization and flexible scheduling (e.g., post-prefill) to further amortize cost. The attention scores re-computation cost for TOVA and SnapKV is ignored in this Figure. Benchmarks report the faster of eager or compiled execution, averaged over 30 runs using bfloat16/TF32 on an NVIDIA B200 GPU.

## A.4. Comparison with the learned baseline JudgeQ

We additionally report a comparison against *JudgeQ* (Liu et al., 2026), a concurrent learned eviction method. JudgeQ resembles KVP in that it treats future generation attention as the supervision signal, but differs in two important ways: (i) it learns soft tokens *inside* the LLM via supervised MSE regression and therefore requires full LLM forward passes during training, while KVP trains a head-local scorer fully offline on pre-collected $(K, V)$ traces; and (ii) at inference time JudgeQ couples its scoring to the LLM's prefill, while KVP runs as a drop-in module on cached $(K, V, \text{position})$ features. We re-implemented JudgeQ from the paper (no code was publicly available) and trained it on the same data as KVP on `Qwen2.5-7B-Chat`.

Figures 8a to 8c reports the long-context comparison: on RULER-128K (top left) KVP maintains a clear lead at every cache budget. On LongBench passage retrieval, both English (top right) and Chinese (bottom left), KVP again outperforms JudgeQ across the budget range. Figure 8 reports the short-context comparison on BoolQ; KVP is on par with or above JudgeQ across all budgets despite being trained without any LLM forward pass.

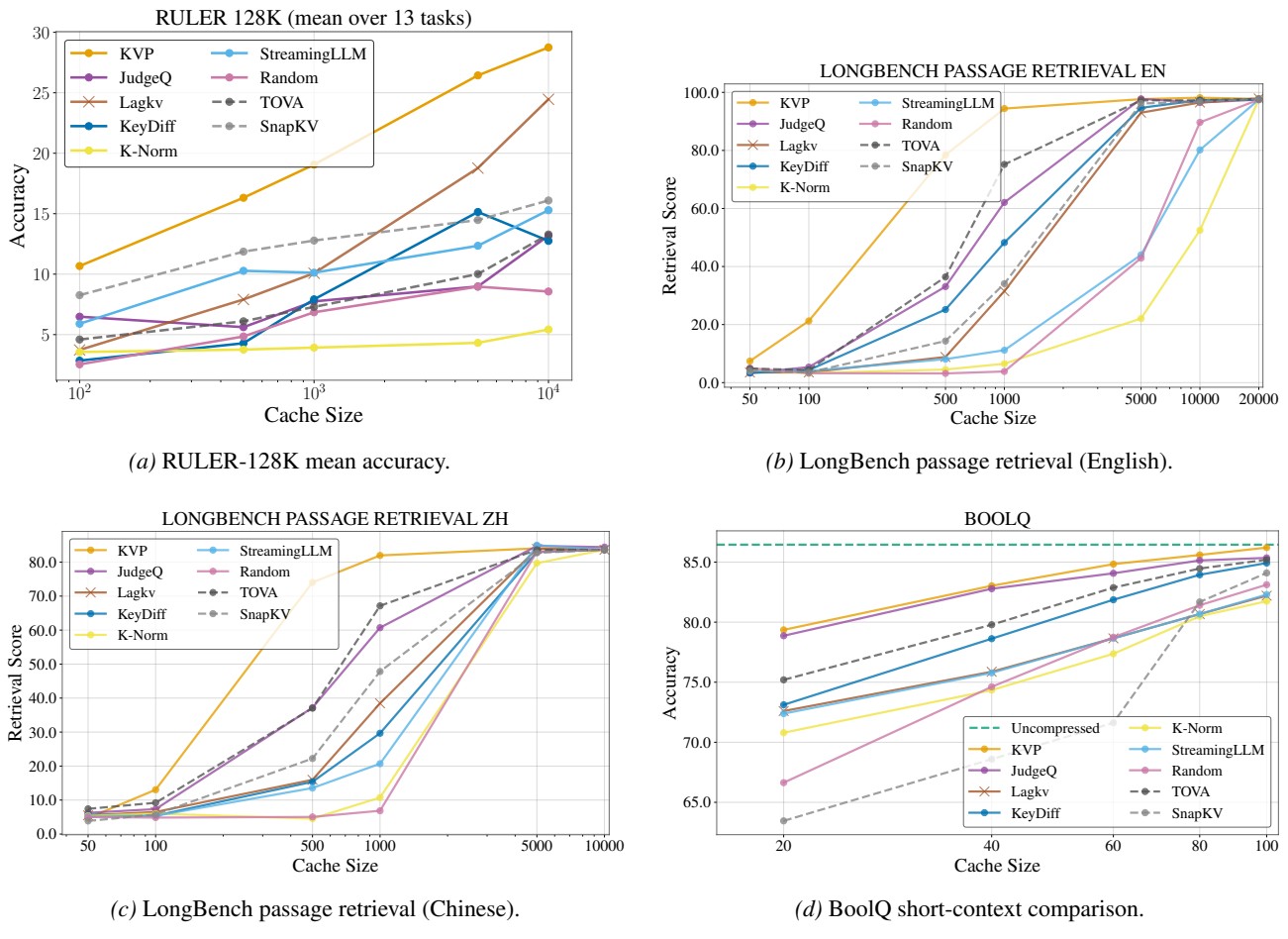

*(a)* RULER-128K mean accuracy.

*(b)* LongBench passage retrieval (English).

*(c)* LongBench passage retrieval (Chinese).

*(d)* BoolQ short-context comparison.

*Figure 8.* Comparison between KVP and the learned baseline JudgeQ. **(a)–(c)** Long-context comparison: KVP outperforms JudgeQ at every cache budget across all three long-context benchmarks. **(d)** Short-context comparison on BoolQ: KVP is on par with or above JudgeQ at every cache budget despite training without any LLM forward pass.

## A.5. Why supervised ranking fails: per-head importance heterogeneity

The main-text ablation (Figure 6b) showed that magnitude-sensitive supervised losses devote capacity to the few attention sinks that dominate the loss. Here we show why that failure is not fixable by tuning: the shape of the per-head importance distribution varies so strongly across heads that no single supervised formulation can be tuned to match all of them.

Figure 9a plots the oracle importance distribution (log scale) for eight representative heads on the same input. The shapes

span the full range from sharp-Zipfian (a few tokens carry essentially all the mass) to near-flat (mass spread across tens of tokens). A fixed magnitude-sensitive loss is calibrated for one of these regimes and fails on the others.

Figure 9b zooms in on a single head (L10.H0). Each bar is a token, positioned at its oracle rank (x) and its oracle attention on a log y-axis, and colored by its *normalized rank error* (green = correctly placed, red = badly mis-ranked). The top panel shows KVP (RL); the bottom panel shows pointwise regression on oracle importance. Regression's red/orange bars concentrate on the *right tail* — the many unimportant tokens. This is the classical overfitting signature of magnitude-sensitive losses on heavy-tailed targets: most of the loss is paid by the left-tail outliers, so the optimizer leaves the right-tail ordering approximately random.

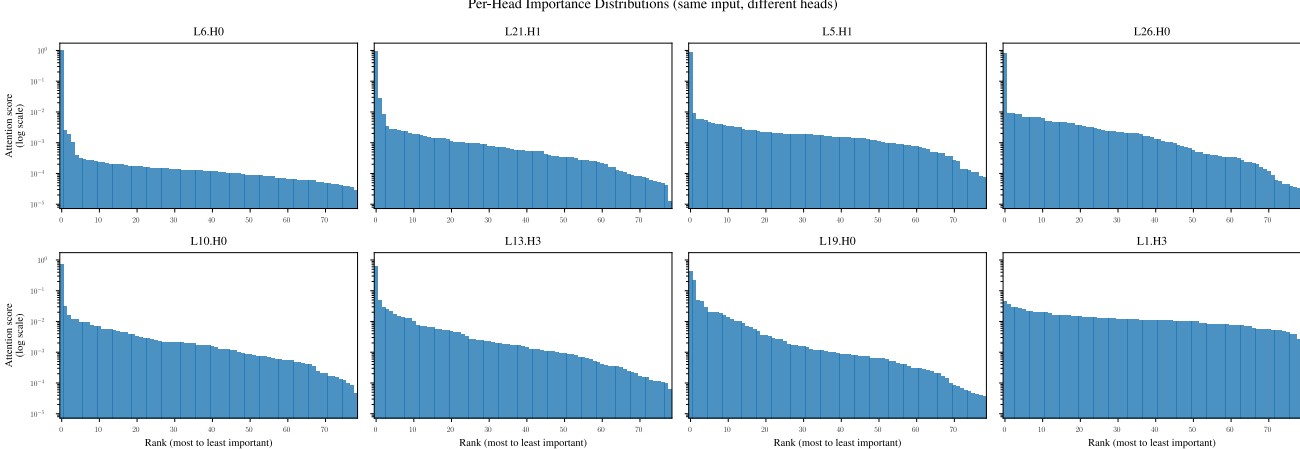

*(a)* Per-head oracle importance distributions on the same input, for eight representative heads. The distribution shape ranges from sharp-Zipfian (e.g. L6.H0, L10.H0) to near-flat (e.g. L1.H3).

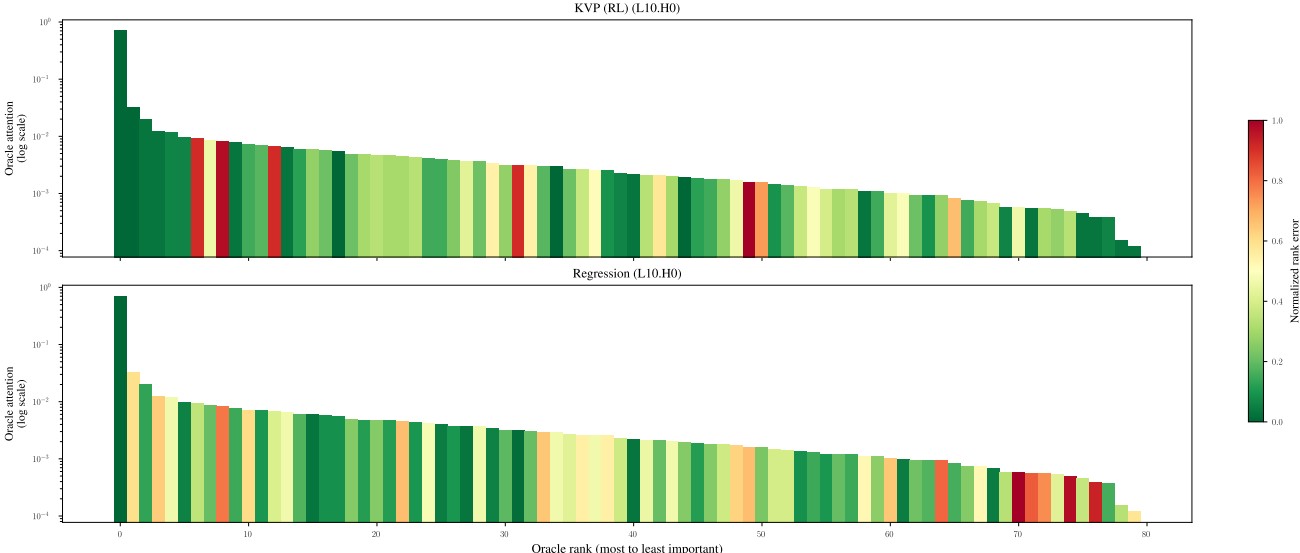

*(b)* For head L10.H0: oracle attention (log) vs. oracle rank, bars colored by normalized rank error (green = correct, red = incorrect). Top: KVP (RL). Bottom: supervised Regression, whose errors concentrate on the right tail.

*Figure 9.* Why supervised ranking under-performs on heavy-tailed attention targets. (a) Per-head oracle importance distributions vary widely, so a single magnitude-sensitive loss cannot be tuned to match all heads. (b) On a representative head, pointwise Regression mis-ranks the right tail; KVP (RL) does not exhibit this bias.

## A.6. Do the per-head agents learn distinct rankings?

A natural question for the per-head architecture is whether the 112 agents converge to similar rankings, in which case a single shared policy would suffice. We show here that the agents in fact learn markedly different rankings for the same input, supporting the per-head design.

We report three complementary views, computed on a qualitative sample from the OASST2 test set using all 112 heads of Qwen2.5-7B (Figures 10 to 12). Figure 11 visualizes the full rank matrix (heads × predicted rank) colored by original token position: a smooth color gradient would indicate a near-identity ranking replicated across heads, whereas the observed disrupted gradients reveal head-specific reordering, with heads converging only on a small early-position band (the attention-sink pattern common to this architecture, recovered without being built in). Figure 10 quantifies this per position: the top panel plots the mean rank with a $\pm 1\sigma$ band across 112 heads, and the bottom panel plots the rank variance per position, which is consistently large (150–500). The **mean rank variance per token position is 294.8**, confirming substantial disagreement. Figure 12 shows the pairwise Spearman rank correlation between heads: heads in the mid-layers (roughly L8–L15) are noticeably more correlated with each other, consistent with their role in information aggregation, while early and late layers are more independent.

We interpret this variance as by design: different heads serve different functional roles (attention sinks, local syntactic patterns, global semantic retrieval), and the per-head agents specialize accordingly. Explicitly modeling cross-head dependencies, e.g. through a shared backbone with head-specific heads or through joint training, is left as future work (see Limitations).

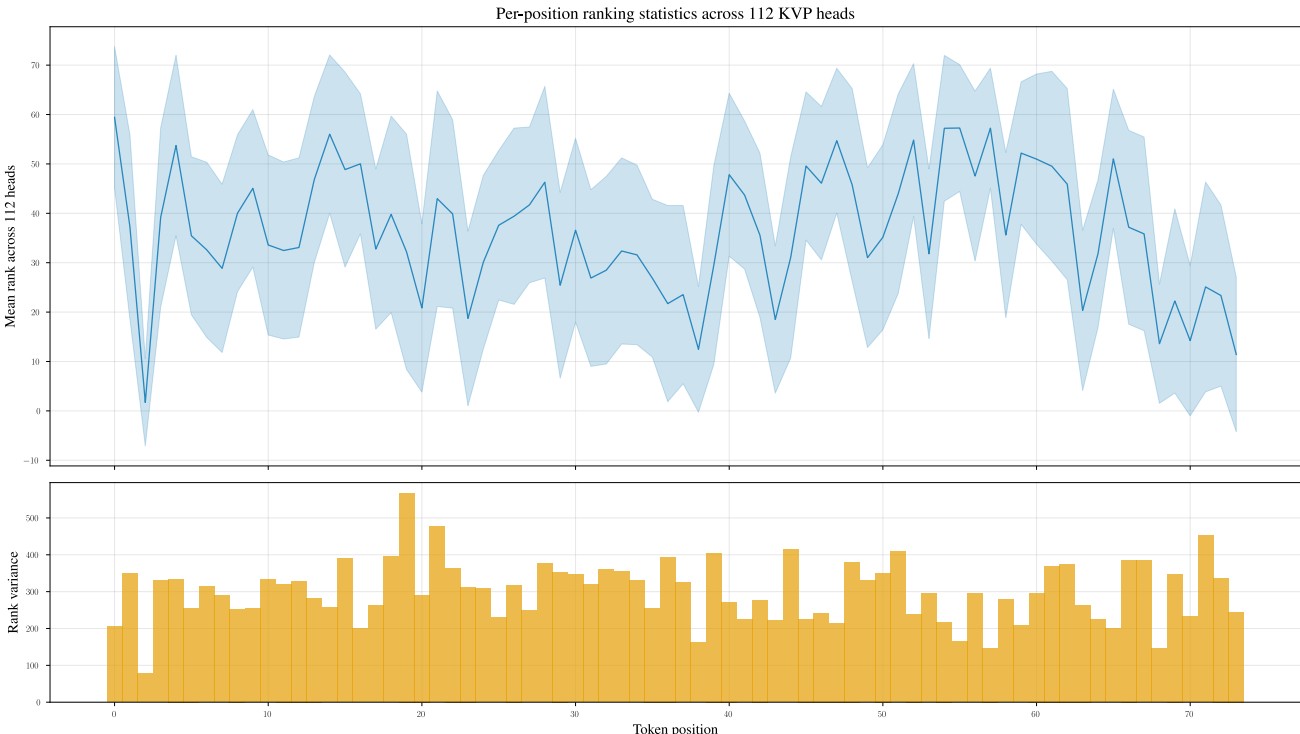

*Figure 10.* Top: mean rank $\pm 1\sigma$ across heads versus token position. Bottom: rank variance per position (mean 294.8 across all positions and heads).

Token Rankings by KVP Agent (112 heads, raw agent scores)

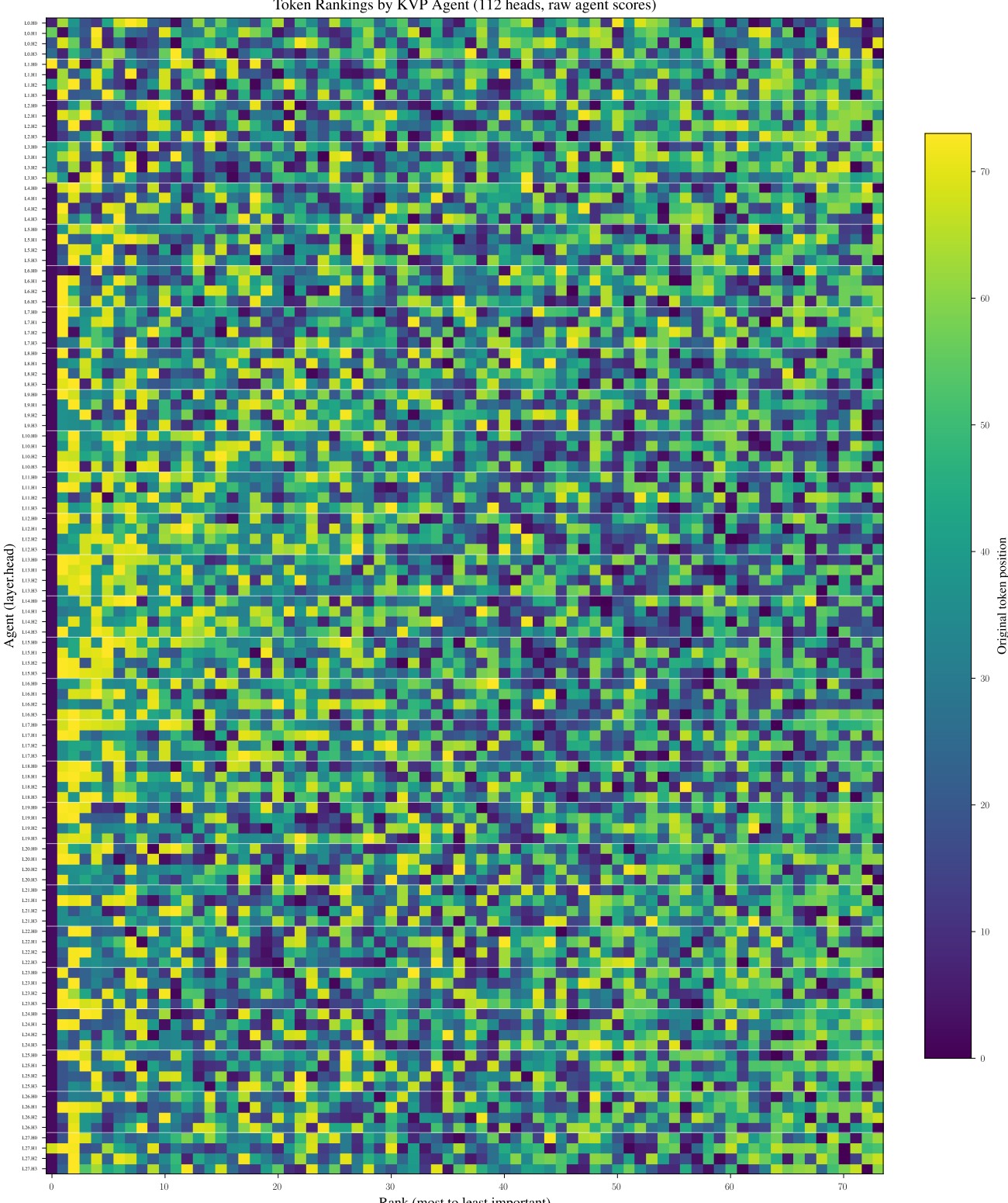

*Figure 11.* Token rankings by each of the 112 KVP agents (rows, ordered L0.H0 → L27.H3), colored by the original token position. Blue stripe in the leftmost columns reflects the attention-sink pattern recovered by many heads (the earliest token positions are colored blue under this colormap and are consistently ranked first); elsewhere rankings disagree strongly.

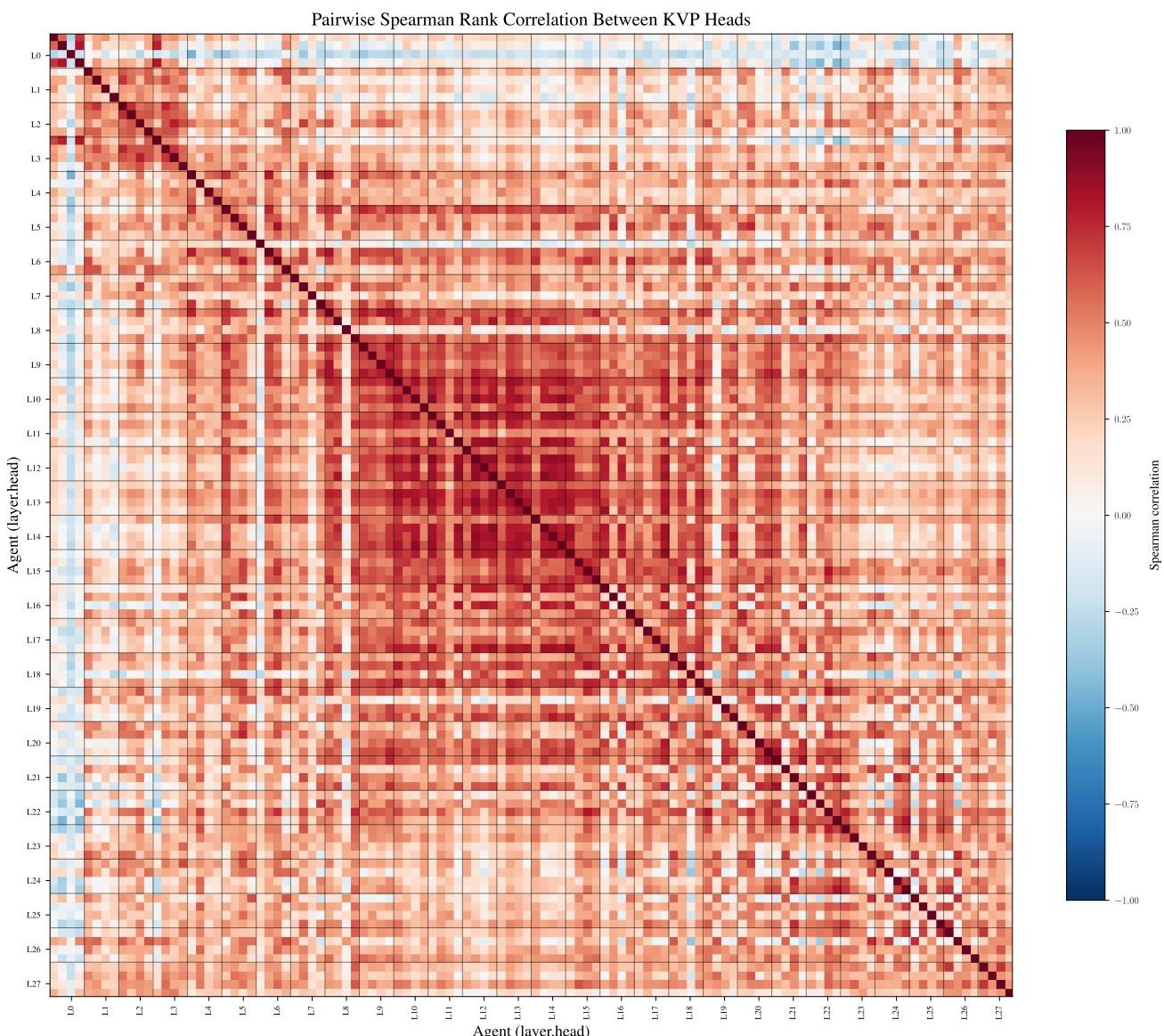

*Figure 12.* Pairwise Spearman rank correlation between KVP heads. Darker = higher correlation. Mid-layers (L8–L15) correlate more strongly; early and late layers are more independent.

## A.7. Additional results

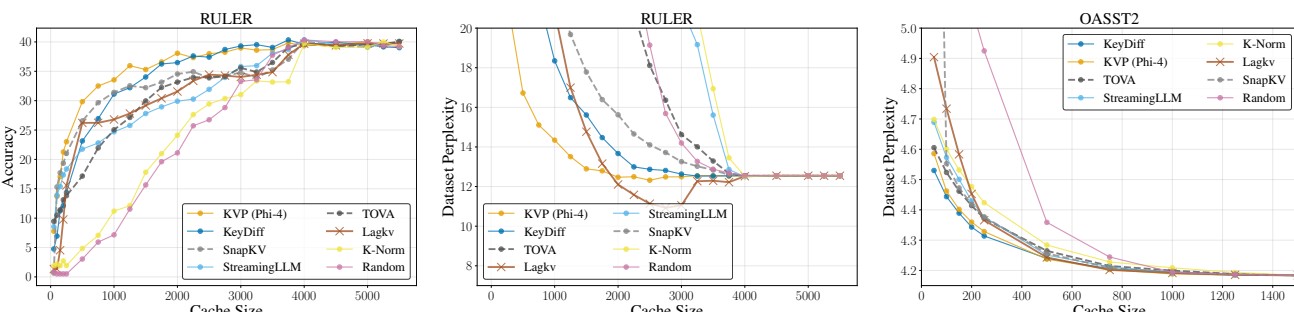

*Figure 13.* Evaluation of our KVP eviction strategy on the `Phi-4 14B` model. The results demonstrate that our learning framework generalizes effectively to a different model architecture. While KVP's performance remains consistently strong, the relative performance of heuristic baselines changes significantly between Qwen 2.5 and Phi-4. This provides strong evidence that their effectiveness is model-dependent, unlike our learned approach. **(left)** KVP achieves the high accuracy on the RULER benchmark, successfully adapting its learned policy where attention-based heuristics (SnapKV, TOVA) fail due to the task's nature. **(center)** Perplexity on the RULER test set, where KVP consistently outperforms baselines. **(right)** Perplexity on the OASST2-4k test set, confirming KVP's good performance.

| Future Attention Importance | KeyDiff | KNorm |
|---|---|---|
| I'm planning a trip to Rome next month. I want to visit the Colosseum, the Vatican Museums, and the Trevi Fountain. What's the best order to visit these attractions to minimize travel time? <\|im_end\|>Also, should I book tickets in advance for these places?<\|im_end\|> | I'm planning a trip to Rome next month. I want to visit the Colosseum, the Vatican Museums, and the Trevi Fountain. What's the best order to visit these attractions to minimize travel time? <\|im_end\|>Also, should I book tickets in advance for these places?<\|im_end\|> | I'm planning a trip to Rome next month. I want to visit the Colosseum, the Vatican Museums, and the Trevi Fountain. What's the best order to visit these attractions to minimize travel time? <\|im_end\|>Also, should I book tickets in advance for these places?<\|im_end\|> |

| KVP | LagKV | Random |
|---|---|---|
| I'm planning a trip to Rome next month. I want to visit the Colosseum, the Vatican Museums, and the Trevi Fountain. What's the best order to visit these attractions to minimize travel time? <\|im_end\|>Also, should I book tickets in advance for these places?<\|im_end\|> | I'm planning a trip to Rome next month. I want to visit the Colosseum, the Vatican Museums, and the Trevi Fountain. What's the best order to visit these attractions to minimize travel time? <\|im_end\|>Also, should I book tickets in advance for these places?<\|im_end\|> | I'm planning a trip to Rome next month. I want to visit the Colosseum, the Vatican Museums, and the Trevi Fountain. What's the best order to visit these attractions to minimize travel time? <\|im_end\|>Also, should I book tickets in advance for these places?<\|im_end\|> |

| SnapKV | StreamingLLM | TOVA |
|---|---|---|
| I'm planning a trip to Rome next month. I want to visit the Colosseum, the Vatican Museums, and the Trevi Fountain. What's the best order to visit these attractions to minimize travel time? <\|im_end\|>Also, should I book tickets in advance for these places?<\|im_end\|> | I'm planning a trip to Rome next month. I want to visit the Colosseum, the Vatican Museums, and the Trevi Fountain. What's the best order to visit these attractions to minimize travel time? <\|im_end\|>Also, should I book tickets in advance for these places?<\|im_end\|> | I'm planning a trip to Rome next month. I want to visit the Colosseum, the Vatican Museums, and the Trevi Fountain. What's the best order to visit these attractions to minimize travel time? <\|im_end\|>Also, should I book tickets in advance for these places?<\|im_end\|> |

*Figure 14.* All strategies compared on the qualitative example shown in Figure 1. The attention scores considered are from layer 12 head 0. Another qualitative example in Figure 19 and a failure case in Figure 15

### Future Attention Importance

If a store offers a 20% discount on a jacket and the sale price is $80, what was the original price? Please calculate the answer and let me know what you get.<|im_end|>The original price was $100. Since the store took 20% off, the sale price represents the remaining 80% of the original price (100% - 20% = 80%). To find the full price (100%), you divide the sale price by 0.80.<|im_end|>

### KeyDiff

If a store offers a 20% discount on a jacket and the sale price is $80, what was the original price? Please calculate the answer and let me know what you get.<|im_end|>The original price was $100. Since the store took 20% off, the sale price represents the remaining 80% of the original price (100% - 20% = 80%). To find the full price (100%), you divide the sale price by 0.80.<|im_end|>

### KNorm

If a store offers a 20% discount on a jacket and the sale price is $80, what was the original price? Please calculate the answer and let me know what you get.<|im_end|>The original price was $100. Since the store took 20% off, the sale price represents the remaining 80% of the original price (100% - 20% = 80%). To find the full price (100%), you divide the sale price by 0.80.<|im_end|>

### KVP

If a store offers a 20% discount on a jacket and the sale price is $80, what was the original price? Please calculate the answer and let me know what you get.<|im_end|>The original price was $100. Since the store took 20% off, the sale price represents the remaining 80% of the original price (100% - 20% = 80%). To find the full price (100%), you divide the sale price by 0.80.<|im_end|>

### LagKV

If a store offers a 20% discount on a jacket and the sale price is $80, what was the original price? Please calculate the answer and let me know what you get.<|im_end|>The original price was $100. Since the store took 20% off, the sale price represents the remaining 80% of the original price (100% - 20% = 80%). To find the full price (100%), you divide the sale price by 0.80.<|im_end|>

### Random

If a store offers a 20% discount on a jacket and the sale price is $80, what was the original price? Please calculate the answer and let me know what you get.<|im_end|>The original price was $100. Since the store took 20% off, the sale price represents the remaining 80% of the original price (100% - 20% = 80%). To find the full price (100%), you divide the sale price by 0.80.<|im_end|>

### SnapKV

If a store offers a 20% discount on a jacket and the sale price is $80, what was the original price? Please calculate the answer and let me know what you get.<|im_end|>The original price was $100. Since the store took 20% off, the sale price represents the remaining 80% of the original price (100% - 20% = 80%). To find the full price (100%), you divide the sale price by 0.80.<|im_end|>

### StreamingLLM

If a store offers a 20% discount on a jacket and the sale price is $80, what was the original price? Please calculate the answer and let me know what you get.<|im_end|>The original price was $100. Since the store took 20% off, the sale price represents the remaining 80% of the original price (100% - 20% = 80%). To find the full price (100%), you divide the sale price by 0.80.<|im_end|>

### TOVA

If a store offers a 20% discount on a jacket and the sale price is $80, what was the original price? Please calculate the answer and let me know what you get.<|im_end|>The original price was $100. Since the store took 20% off, the sale price represents the remaining 80% of the original price (100% - 20% = 80%). To find the full price (100%), you divide the sale price by 0.80.<|im_end|>

*Figure 15.* A qualitative comparison of all strategies, similar to Figure 1. In this particular failure case, the attention scores from layer 10, head 0, show that our proposed KVP strategy struggles to fully capture the ground-truth future attention pattern.

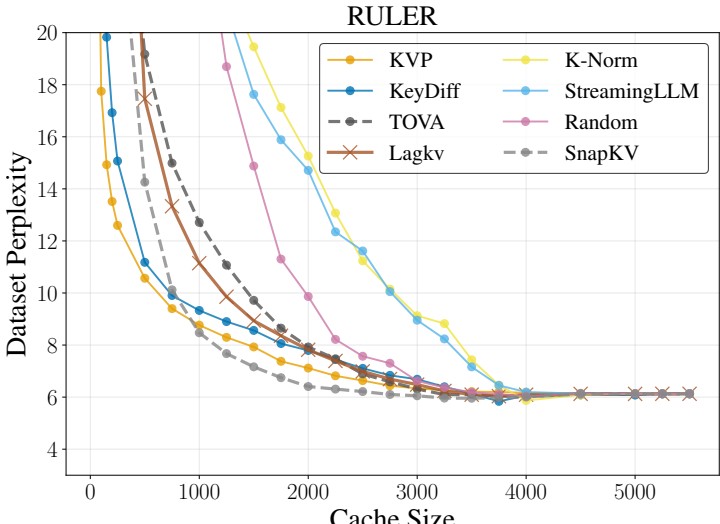

*Figure 16.* Perplexity (PPL) as a function of KV cache size. KVP achieves highly competitive perplexity, performing on par with or better than the leading baselines at most cache sizes and significantly outperforming other methods. This result is particularly notable given that RULER's structure, which includes random sentences preceding a final question, heavily advantages methods that can isolate tokens relevant to that question.

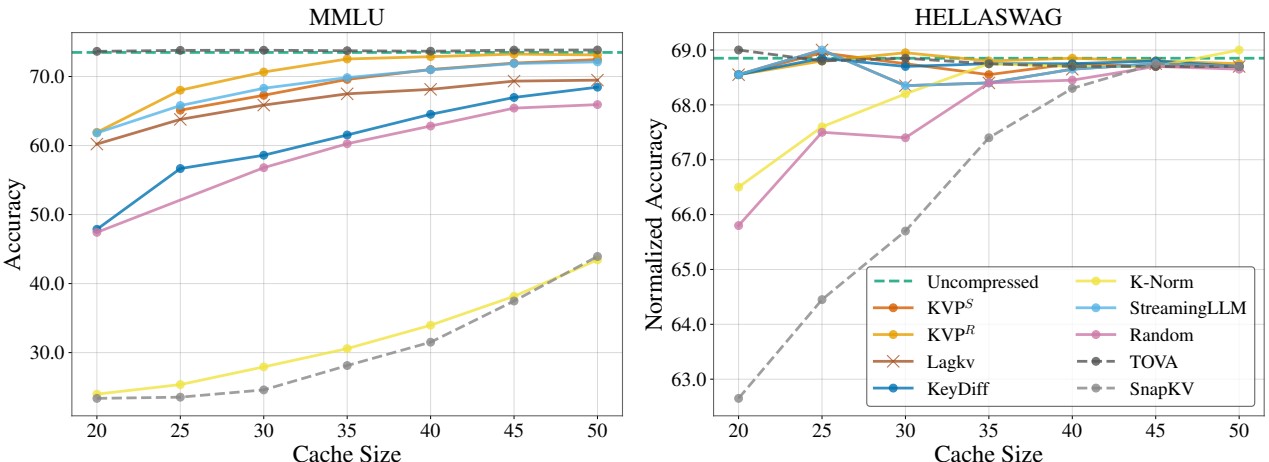

*Figure 17.* (Left) Average test accuracy on MMLU and (Right) average normalized accuracy on Hellaswag as a function of KV cache size. Higher is better.

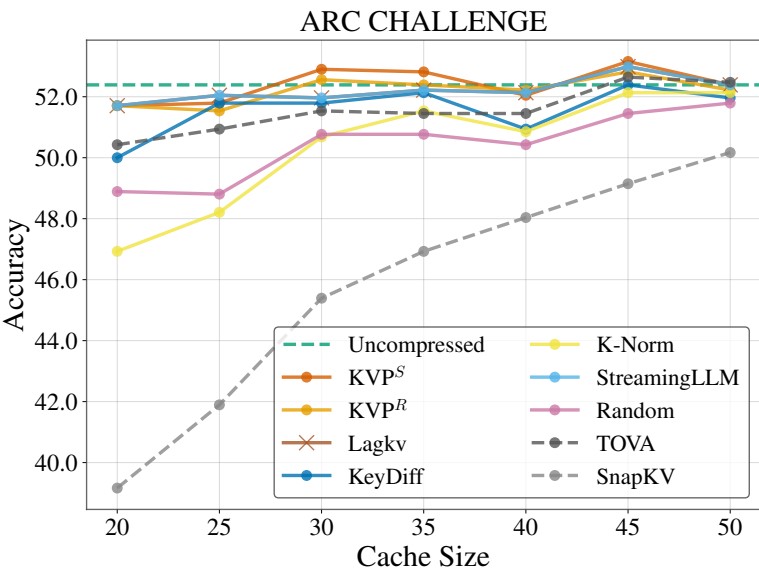

*Figure 18.* Average test accuracy on *ARC-Challenge* as a function of KV cache size. Higher is better. ARC-Challenge probes multi-step science reasoning; both $KVP^R$ and $KVP^S$ remain competitive with the uncompressed model across the budget range.

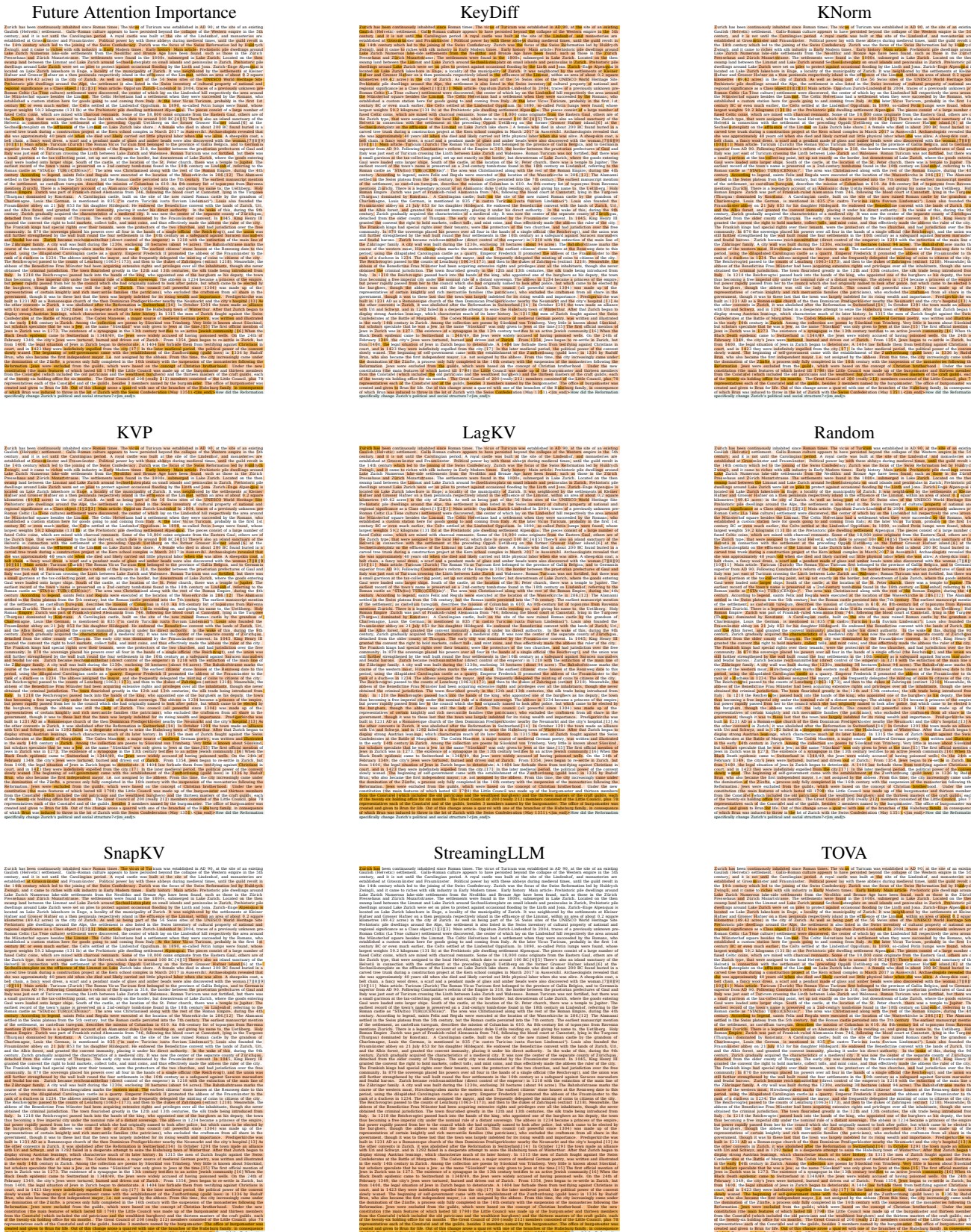

*Figure 19.* All strategies compared on a long qualitative example. The attention scores considered are from layer 19 head 0.

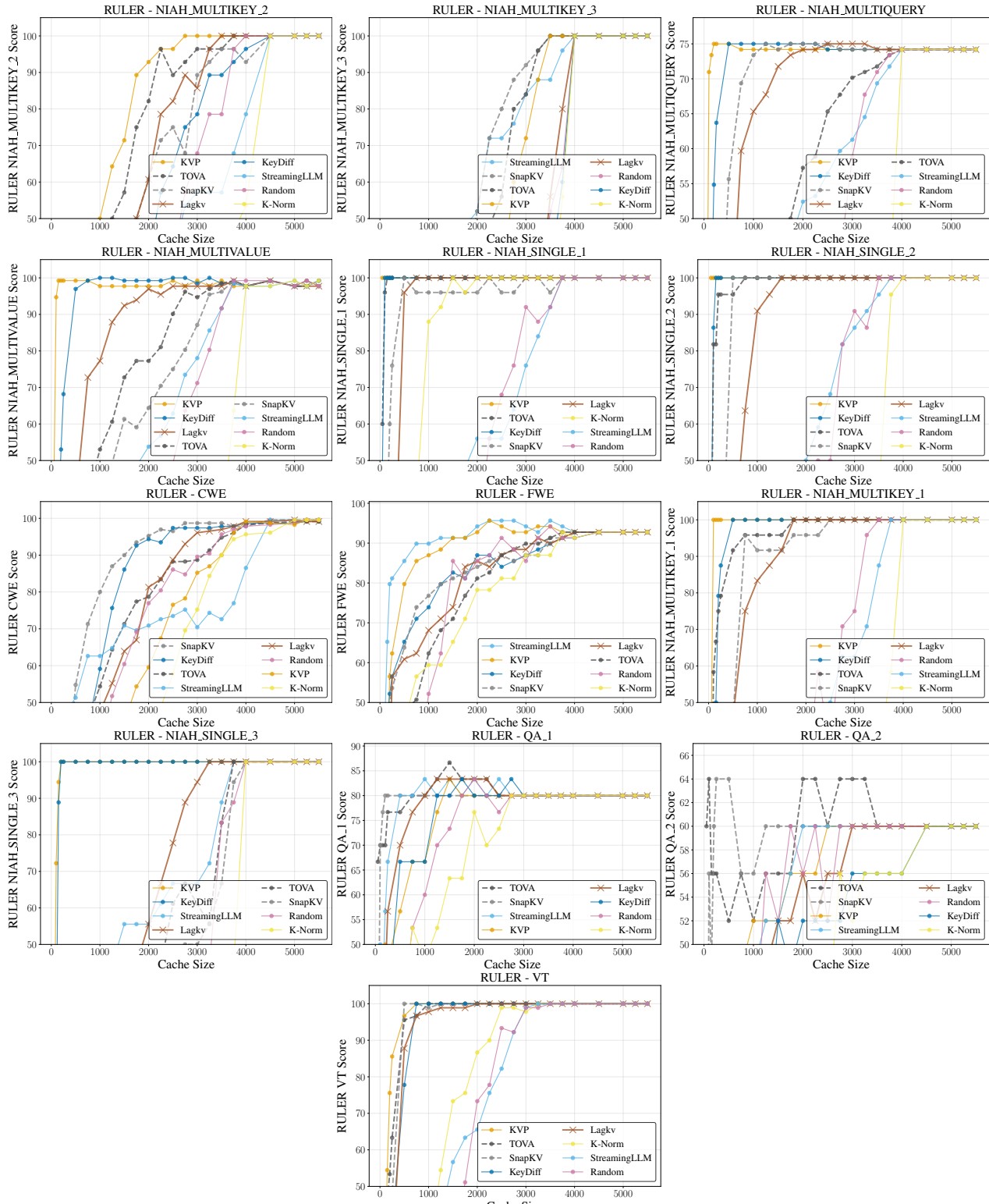

*Figure 20.* Per-task accuracy on all RULER subtasks as a function of absolute KV cache size.

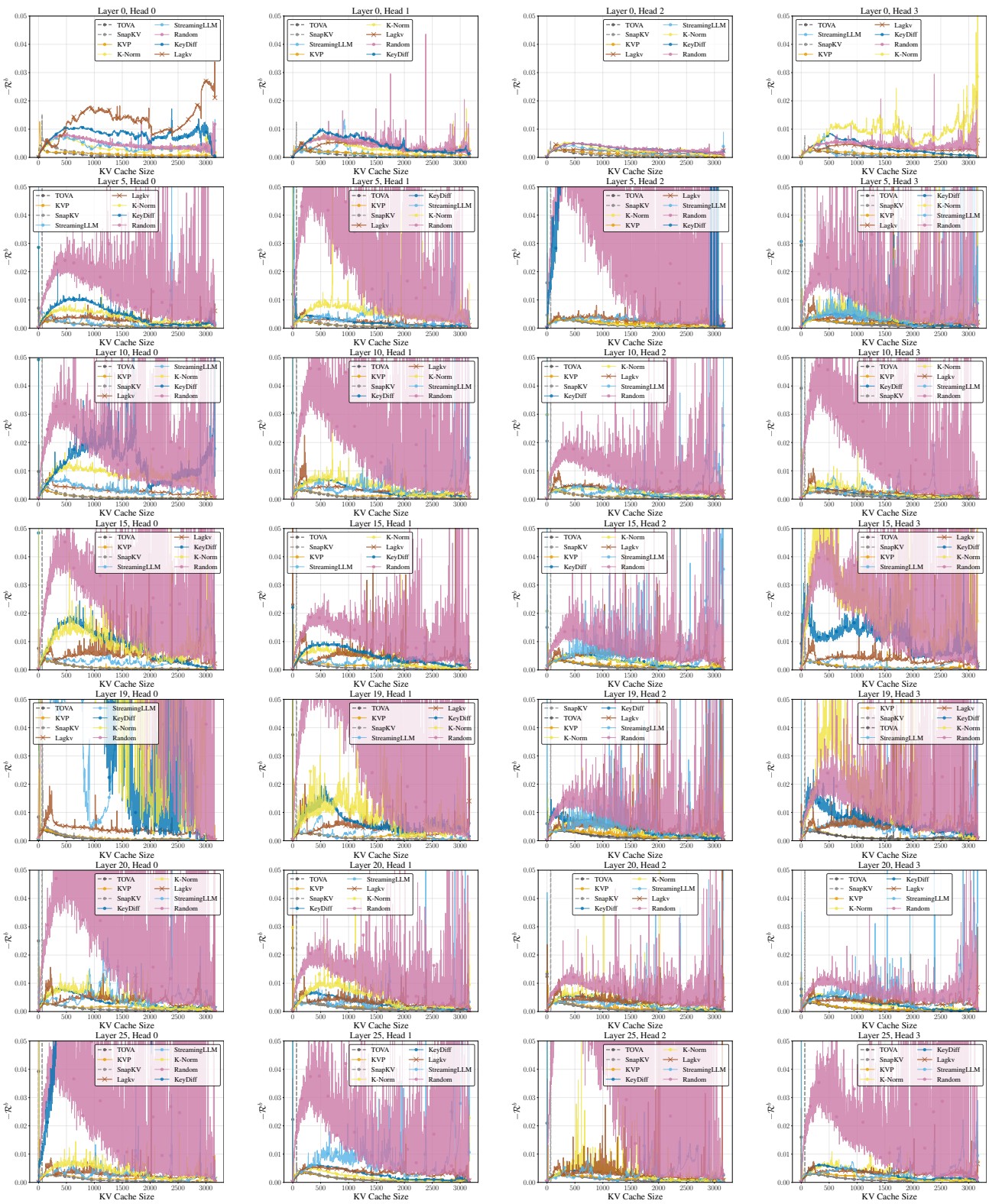

*Figure 21.* Cost $(-\mathcal{R}^b)$ for all strategies across a selection of layers (rows) and all available heads (columns) on the OASST2 test set. The plots show that the relative performance of different strategies varies significantly across heads, highlighting the benefit of learning specialized per-head policies. Lower is better.

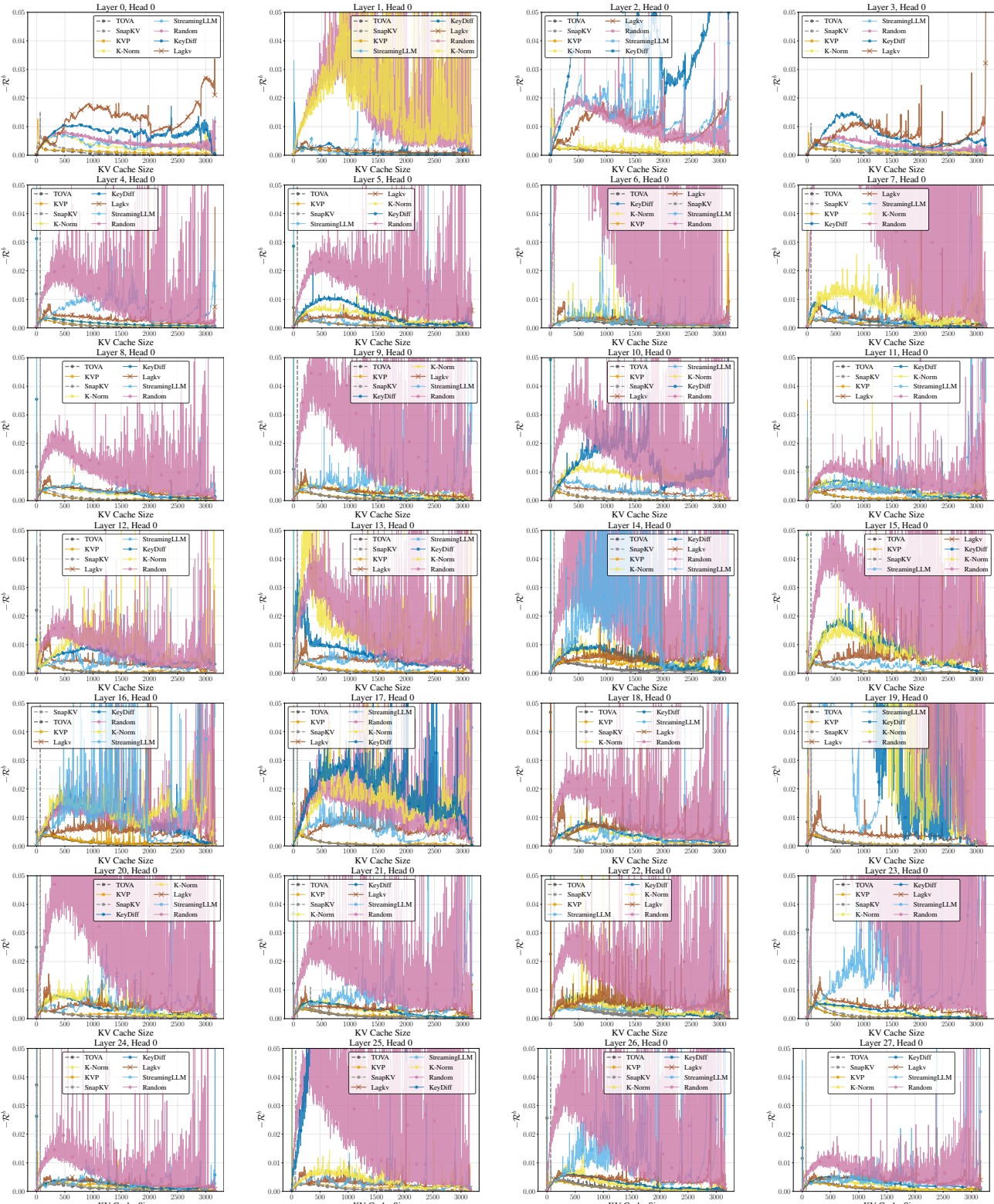

*Figure 22.* Cost ($-\mathcal{R}^b$) for all strategies on head 0 across all 28 layers of the model, evaluated on the OASST2 test set. This visualization shows how the effectiveness of different non attention-aware eviction heuristics changes with model depth, whereas the learned KVP policy remains consistently effective. Lower is better.

