# OpenReview forum: "Learning to Evict from Key-Value Cache"
_ICML.cc/2026/Conference — ICML 2026 regular_

### Official Review · Reviewer_tB21 · 2026-02-21

**Soundness:** 1
**Presentation:** 3
**Significance:** 2
**Originality:** 3
**Overall Recommendation:** 4
**Confidence:** 3

**Summary:**

This paper proposes Key-Value Policy (KVP), a learned KV-cache eviction policy that ranks cached tokens according to “future usefulness,” so that for any cache budget b you keep the top-b tokens and evict the rest. To do this, they train lightweight RL agents, one per MLP head, which produce a score as a function of the keys, values, and position of a token. The training is done offline, where they precompute traces of Q, K, and V, from the base transformer model, then define importance using future attention, and optimize a global reward that sums eviction quality over all budgets. During inference they run each of head’s MLPs on cached KV entries, sort the tokens, then keep top-b entries for a selected cache size.

**Compliance With Llm Reviewing Policy:**

Affirmed.

**Final Justification:**

My justification is written in the rebuttal acknowledgement. My major concerns were addressed, I am putting weak accept because I believe there is more room for a fully defensible paper.

**Key Questions For Authors:**

My questions are posed in the weaknesses subsection of the strengths and weaknesses section.

**Limitations:**

They address limitations only lightly, mostly as “future work,” and they do not have a dedicated “Limitations” section.

**Strengths And Weaknesses:**

Strengths:
- Strong empirical results. KVP often beats attention-free heuristics and competes well with attention-based methods that are given attention scores from the “prefill” stage. Also, limited degradation versus baselines on trained agents with zero-shot on BoolQ, ARC-Challenge, and GovReport.
- They achieve per head specialization, meaning different heads exhibit different important structures.
- It is a practical approach. The policy depends only on KV and positions without depending on query during inference. Also, the overhead is concentrated in prefill with small FLOPs.

Weaknesses:

I am confused about the NP-Hard claim “even for linear rewards” and proposition 1.

- Firstly, you reference the Nemhauser 1978 paper, which does not exactly support what you are saying. That reference would agree that this kind of set-function maximization includes NP-hard problems, but the “even linear rewards” is where you lose me. Unless I am misunderstanding, this is what I am gathering: if you have a bunch of items with scalar weights w_i and you want exactly b items, then you want to maximize a sum over w_i, for i in S, such that |S| = b. This is not “hard”, there is no structure in constraint beyond pick b, the solution is just sort w_i and take the top b (regardless of uniqueness or other assumptions). This is not equivalent to the Nemhauser 1978 paper, where they study a much broader class. I need more clarification, I’m either missing something or the framing is off. If you intended a broader constraint family or a nonlinear true objective (e.g., post-eviction likelihood), you should state that precisely and revise the claim.
- Now regarding the practicality of assumptions. The “nestedness” is strong for general rewards R(s). In many tasks, the best set of size b needs not be a subset of the best set of size b+1 (e.g., think complements or substitutes). However in your actual reward (future-attention-based and additive over tokens), nestedness actually does hold, which then makes the theoretical framing disconnected for this NP-hard motivation. It is looking like you are using a “hard combinatorial selection” story to justify the method, but the reward you actually optimize reduces to sorting scalars, which is not combinatorially hard.

Notation in reward eq (4) and the definition of your objective:
- I think the notation is confusing as written. If “budget b” means you KEEP the top-b tokens, then the evicted set is the suffix sigma_{b+1},…,sigma_n (size n-b). So when you write something like “Rb(sigma_{n−b},…,sigma_n)”, that looks like you are feeding the LAST (b+1) tokens (because n-b to n is b+1 elements), which is neither the kept set (top-b) nor the full evicted set (indices b+1 to n). For example, take n=10 and b=2. Evicted tokens should be rank 3…10 (that is 8 tokens).  Your written suffix of sigma_{n−b},…,sigma_n would give sigma_8,…,sigma_10, which is 3 tokens.
- More alarmingly: The reward normalization is likely sign-inconsistent (could invert optimization if implemented literally). You define per-budget reward as the negative eviction cost: Rb = negative sum of importance of evicted tokens. That makes Rb <= 0. You then say they “normalize the total cost by the cost incurred by an optimal ranking sigma*” and optimize R(sigma)/R(sigma*) and Rb(sigma)/Rb(sigma*). If you literally maximize Rb(σ)/Rb(σ*), you can end up preferring worse policies because both numerator and denominator are negative. So say for example: let optimal cost C*=1, so R*=-1. Let a worse policy have cost C=2, so R=-2. Then normalized would give R/R* = 2, which is larger than 1 and hence maximizing this ratio could push toward larger costs (worser evictions). Either define reward as a saved positive utility, or normalize the cost (not the negative cost), or something like (1-C/C*). At minimum, this requires a clear statement of what is actually maximized in code.

The “global reward across all budgets” collapses to a simple weighted sorting objective.
- To explain: Given the per-token importance as the summation over future attention u_i = sum from j=n+1 to n+f of A(x_i, x_j), the per-budget reward Rb(sigma) is a negative evicted importance. Summing across b yields eq(4) (R = summation over b of Rb) where Rb is given in eq(5) which we can write as Rb = - sum from i = b+1 to n of u_{sigma_i}. For some priority list, fix rank r (the rth position of the permutation). The token at that position is x_{sigma_r}, and it appears in the evicted suffix {sigma_{b+1},…,sigma_n} exactly when r>b (since budget keeps the first b ranks). Because b in {1,…,n-1}, the condition r>b holds for b=1,2,…,r-1 (exactly r-1 budgets). Therefore when you sum Rb across all b, the term u_{sigma_r} is counted (r-1) times. This gives the closed form R(sigma)= - sum from r=1 to n (r-1)u_{sigma_r}. So the global objective is just a weighted sum of the per-token importances, with weights increasing with rank: rank 1 has weight 0, rank 2 has weight 1, rank 3 has weight 2, and so on. This immediately implies the optimal permutation is obtained by sorting tokens by decreasing u_i (largest importance placed earliest). You can see this by a two-item swap: take two tokens a,b with u_a > u_b placed at ranks r<s. Their contribution to the “cost” -R is (r-1)u_a + (s-1)u_b. If you swap them it becomes (r-1)u_b + (s-1)u_a. The differences is thus ((s−1)−(r−1))(u_a−u_b)=(s−r)(u_a−u_b)>0, meaning the swap strictly increases the cost and decreases the reward. So the better arrangement is to keep the larger u at the smaller rank. In short, under your reward proxy, the “combinatorial” selection problem reduces to a deterministic ranking-by-u problem, which raises the question of whether policy-gradient RL is necessary versus a simpler supervised objective that directly learns to predict u_i and then sorts. You have evidence that your offline RL works and that one differentiable-sorting supervised baseline fails, but the claim ‘RL is necessary’ is premature without regression/pairwise/listwise supervised baselines and a stronger hyperparameter/seed sweep.

Other concerns:
- Context lengths in benchmarks are relatively modest (4-5k KV Cache size in OASST2). The paper should include at least one benchmark regime that truly stresses very long contexts.
- Practicality of the size of storing the full Q, K, V for entire sequences (memory can blow up quickly). But you mention this being lightweight. Can you report an end-to-end system result where the cache is actually compacted (not just masked), including memory usage?

---

> ### Author Rebuttal · Authors · 2026-03-31
>
> We thank Reviewer for their thorough and technically detailed review. Several of the reviewer's observations actually support the method: the reduction to weighted sorting confirms our modeling choices convert a generally hard problem into a tractable one. We verified that the implementation is correct; the core issues are presentational.
>
> ## 1. NP-Hard Claim, Nestedness, and Notation (Sections 3, Eq. 4)
>
> We agree that the NP-Hard claim is confusing and unnecessary. We originally included it to acknowledge that subset selection is NP-hard even in the linear case (e.g., with linear rewards and linear costs as in Knapsack). We will remove this claim and revise the text as follows:
>
> "...Although this selection problem is NP–hard even for linear rewards (Nemhauser et al., 1978), it becomes structured under two mild conditions:..." to "...This selection problem becomes structured under two mild conditions:..."
>
> We did not mean to claim we are solving an NP-hard problem in our specific formulation. Rather, KV-cache selection is a "hard combinatorial selection" problem in the general case, and we specifically defined a setting (nestedness and uniqueness) where the problem becomes tractable. Our reward function is designed to satisfy the "nestedness" property, which reduces the objective to a sorting solution that we then utilize to learn our policies. We will revise Section 3 to make these distinctions and the notation in Eq. 4 clearer.
>
> ## 2. Sign Inconsistency in Reward Normalization
>
> The reviewer is correct that the paper's description is inconsistent. The implementation correctly normalizes positive costs (C(σ)/C(σ*) ≥ 1) and negates once to form the reward (−C(σ)/C(σ*)), so the RL objective minimizes the cost ratio as intended. We will rewrite the description to match the implementation.
>
> ## 3. Global Reward Collapses to Weighted Sorting; Is RL Necessary?
>
> Prompted by the reviewer's feedback, we have now formally implemented the full suite of requested supervised baselines (Pointwise (Regression on u_i), Pairwise (RankNet), and Listwise (ListNet)) using the exact same setting, architecture, data, and formulation as our RL agent. Per-budget eviction cost (−R^b) on OASST2 test set:
>
> | Cache Size | Ours (RL) | Regression | Pairwise | Listwise |
> | ---: | ---: | ---: | ---: | ---: |
> | 250 | 0.0025 | 0.0053 | 0.0053 | 0.0054 |
> | 500 | 0.0015 | 0.0066 | 0.0066 | 0.0070 |
> | 1000 | 0.0009 | 0.0062 | 0.0052 | 0.0052 |
> | 2000 | 0.0004 | 0.0030 | 0.0031 | 0.0029 |
> | 3000 | 0.0003 | 0.0014 | 0.0036 | 0.0040 |
> | **AUC** | 2.86 | 14.50 | 13.82 | 13.71 |
>
> Please see [Figure 4](https://anonymous.4open.science/api/repo/icml-rebuttal-2026-65E6/file/ICML%20Rebuttal%202026.pdf?v=31d0204e).
>
> All three supervised baselines exhibit the same failure mode, driven by the heavy-tailed distribution of LLM attention: because their losses weight errors by magnitude, they allocate nearly all learning capacity to matching the massive attention sinks, performing well only at extremely small cache sizes but failing to learn the relative ordering of the remaining tokens.
> Conversely, purely ordinal methods like soft-rank (already shown in Figure 5b) that do not account for attention magnitude fail because they assign equal error cost to swapping important tokens and the vast majority of unimportant ones. By directly optimizing the AUC of the eviction cost via policy gradients, the RL agent naturally penalizes swaps proportionally to their actual cost across all budgets, sidestepping both failure modes.
>
> This finding is consistent with JudgeQ (Liu et al., 2025), a recent learned eviction method that trains soft tokens on generation attention via supervised MSE regression. We reimplemented JudgeQ following the paper (no code was publicly available) and trained it on our data. KVP outperforms this reimplementation across all settings (see Section 4), providing further evidence that supervised regression on this signal underperforms RL ranking optimization.
>
> ## 4. Context Lengths
> We now include evaluation on LongBench passage retrieval and RULER at 128K. We also compare against JudgeQ (Liu et al., 2025), a learned eviction method.
> Please refer to [Figures 5, 6 and 7](https://anonymous.4open.science/api/repo/icml-rebuttal-2026-65E6/file/ICML%20Rebuttal%202026.pdf?v=31d0204e):
>
> ## 5. End-to-End Memory
>
> At inference time, KVP stores nothing beyond the standard KV cache, both GPU memory and storage are identical to any other eviction method at the same cache budget. The Q/K/V traces are only needed during offline training and reside on disk. Since each agent is fully independent, traces can be trivially partitioned: only the relevant layer/head data needs to be downloaded for each agent, enabling parallelization across devices with minimal memory footprint. We note that our long-context experiments (RULER 128K, LongBench) use actual KV cache eviction with physical tensor compaction, confirming practical feasibility beyond masked evaluation.

---

> > ### Author Rebuttal · Reviewer_tB21 · 2026-04-03
> >
> > Thank you, you addressed most my concerns. I will change my score to a 4. I believe a full accept requires additional tweaks that entirely put an end to all skpetism. In other words, you could also show tryingreasonable “fixes” to supervised training under heavy-tailed labels (Huber/quantile/log transform/clipping; cost-sensitive pairwise/listwise where weights depend on |u_i - u_j∣ or on the same per-budget weights the RL objective induces; LambdaRank-style weighting). Right now the explanation (“supervised losses chase attention sinks; ordinal soft-rank ignores magnitude”) is plausible, but it needs either (i) ablation evidence that supervised models indeed overfit sinks (e.g., errors concentrated on non-sink tokens / poor ordering among the middle mass), or (ii) supervised variants explicitly designed to target their AUC-over-budgets objective. Otherwise one can still say “you compared to a few standard losses, but not the best cost-sensitive supervised formulation that matches your objective."
> >
> > Finally, adding 128K eval and claiming physical tensor compaction directly addresses the “masked vs real eviction” concern, but you still don’t provide hard reproducibility numbers for the offline traces (total disk footprint, I/O, training time) and inference overhead (time to score/sort/compact at 128K). The “memory is identical at the same budget” sentence is true but also kind of redundant. What matters is the extra overhead and whether the full pipeline is practical.

---

> > > ### Author Response · Authors · 2026-04-07
> > >
> > > We thank the reviewer for the constructive follow-up and the concrete suggestions. We address both remaining points:
> > >
> > > **1. Ablation evidence that supervised models overfit sinks**
> > >
> > > We provide two pieces of evidence:
> > >
> > > **(a) Per-head importance distributions are highly heterogeneous.** [Figure 1](https://anonymous.4open.science/api/repo/icml-rebuttal-11FF/file/image.png?v=c192b60f) shows the oracle (future attention) importance distribution across 8 representative heads for the same input. Different heads exhibit very different distribution shapes. While we agree with the reviewer that it is possible to tune and tweak a supervised method to fit a specific distribution, it is challenging to devise a single supervised formulation that works well across all types of distributions encountered in the heads. In contrast, our RL formulation naturally adapts to each head's distribution without requiring any per-head tuning or loss design, as shown in Figure 14 and 15 in the Appendix.
> > >
> > > **(b) Supervised ranking errors concentrate on the right side of the distribution.** [Figure 2](https://anonymous.4open.science/api/repo/icml-rebuttal-11FF/file/image2.png?v=e0e36d37) shows, for a specific head, the oracle importance of each token sorted by ground truth rank (most to least important), with bars colored by the ranking error of each method. The Regression baseline correctly ranks the leftmost tokens (attention sinks) but concentrates most of its errors on the right side, the less important tokens.  This confirms that supervised losses learn to match the dominant sinks but fail on the ordering among the remaining tokens.
> > >
> > > **2. Reproducibility numbers for offline traces and inference overhead**
> > >
> > > - **Offline traces**: Processing the entire OASST2 dataset (4,649 samples, 28 layers × 4 KV heads) took less than two hours using 7 nodes with 8 GPUs each, since the computation is embarrassingly parallel. Each node processes data independently, so the full dataset never resides on a single machine. The full dataset occupies ~1.2 TB on cluster storage; a single (layer, head) partition is ~11 GB.
> > > - **Training time**: Each agent requires only its ~11 GB partition and trains in ~30 minutes on 8 H100s (5 epochs). All 112 agents train fully in parallel, making the entire pipeline, from trace generation to agent training, easily parallelizable and scalable.
> > > - **Inference overhead**: As reported in Figure 6 and Appendix A.3, the per-head scoring overhead, the component unique to KVP, adds 0.71ms per layer at 10K context, 570× lower than the LLM prefill (404ms). The FLOPs overhead is ~1% of prefill cost, with zero additional cost during autoregressive generation. Sorting and physical tensor compaction are O(n log n) and O(n) respectively and are shared by all eviction methods. We note that these numbers isolate the scoring component; our current codebase does not yet include an optimized end-to-end inference pipeline, so we cannot report full system wall-clock times at 128K.

---

### Official Review · Reviewer_774D · 2026-03-10

**Soundness:** 3
**Presentation:** 3
**Significance:** 3
**Originality:** 3
**Overall Recommendation:** 5
**Confidence:** 2

**Summary:**

The paper addresses the memory bottleneck of maintaining a KV cache for LLM inference via an eviction methodology. The authors reframe the eviction decision problem as a RL ranking problem. They train lightweight RL agents to rank tokens by their predicted future utility using only the information available in the cache, sort tokens based on the predicted performance and retaining KV values for only the top tokens.

**Compliance With Llm Reviewing Policy:**

Affirmed.

**Final Justification:**

The reviewers have answered all my questions and the work provides a good justification for all my comments.

**Key Questions For Authors:**

1. Since tokens identified as "important" vary by head, how much variance was observed in the rankings produced by different heads for the same input sequence?

**Limitations:**

yes

**Strengths And Weaknesses:**

Strengths:
1. Reframing KV eviction as a ranking task rather than a binary classification (keep/evict) is a significant conceptual shift.
2. Because the policy relies only on static features already in the cache, it is computationally efficient.
3. The empirical results showcase the methodology yielding significant gain while it is agile and has significant room for improvement as it opens a new path to memory management for KV caching.

Weakness:
1. While the agents are "lightweight," the paper requires training a distinct agent for every head in the LLM. This cost can significantly add up for large LLMs with dozens of of attention heads. Also, the approach adds significant retraining cost towards testing any architectural changes.
2. The per head approach can lead to disagreement across heads in terms of which tokens should be preserver. While the current approach removes complete sync among heads for such decisions, one would imagine that this eviction would be a holistic decision not a per head one. Can the authors provide some justification as to why this aspect leads to better results or a possible comparison to what keeping sync among heads would yield?

---

> ### Author Rebuttal · Authors · 2026-03-31
>
> We thank the Reviewer for their positive assessment and thoughtful questions. We address each concern below.
>
> ## 1. Per-Head Training Cost
>
> We appreciate this concern. However, the per-head architecture is extremely parallelizable: all 112 agents are independent and can be trained concurrently across GPUs with no communication overhead. In our implementation, agents can be distributed across devices trivially. This means that scaling to larger models (e.g., with more layers or heads) increases total compute but not wall-clock time, given sufficient parallelism.
>
> Concretely, the total one-time training cost is negligible with enough parallelism and significantly less than the fine-tuning cost:
>
> - **Parameter count**: Each agent is a 2-layer MLP with ~650K parameters. For Qwen2.5-7B-Chat with 4 KV heads across 28 layers, this yields 112 agents totaling ~73M parameters, roughly 1% of the base model.
> - **Training time**: <30 minutes on 8 NVIDIA H100 GPUs.
> - **Checkpoint size**: 2.6MB per agent.
>
> ## 2. Cross-Head Coordination and Ranking Variance
>
> This is an excellent question. We provide three new analyses on a qualitative sample from OASST2, considering all 112 heads:
>
> **Ranking visualization** (heads vs. predicted rank, colored by original token position). A smooth color gradient indicates an identity-like ranking; disrupted gradients reveal head-specific reordering. The visualization confirms that different heads learn markedly different rankings for the same sequence. See [Figure 3](https://anonymous.4open.science/api/repo/icml-rebuttal-2026-65E6/file/ICML%20Rebuttal%202026.pdf?v=31d0204e).
>
> **Mean rank and variance across heads** (as a function of token position). The mean rank variance across all 112 heads is 294.8, confirming substantial disagreement. See [Figure 2](https://anonymous.4open.science/api/repo/icml-rebuttal-2026-65E6/file/ICML%20Rebuttal%202026.pdf?v=31d0204e).
>
> **Pairwise Spearman rank correlation between heads.** Interestingly, heads in mid-layers show higher correlation, while early and late layers are more independent. See [Figure 1](https://anonymous.4open.science/api/repo/icml-rebuttal-2026-65E6/file/ICML%20Rebuttal%202026.pdf?v=31d0204e).
>
> This variance is by design, different heads serve fundamentally different functions (e.g., attention sinks, local syntactic patterns, global semantic retrieval). Figures 14 and 15 in the Appendix further confirm this. The per-head independence in KVP allows each head to retain exactly the tokens it needs. We discuss cross-head coordination as a promising future direction in Section 5.
>
> ## 3. Additional Experiments
>
> During the rebuttal period, we have conducted additional experiments that further strengthen the paper.
>
> **Learned baseline (JudgeQ).** We compare against JudgeQ (Liu et al., 2025; reimplemented following the paper; no public code available), a learned eviction method that trains soft tokens to approximate the same generation attention signal as KVP, requiring full LLM forward passes during training. KVP outperforms JudgeQ across all settings while being fully offline.
>
> **Long-context evaluation.** We evaluate on RULER at 128K context length and LongBench passage retrieval. Both use a chunked prefill-compress loop, testing KVP under repeated compression.
>
>
> RULER 128k (See [Figure 5](https://anonymous.4open.science/api/repo/icml-rebuttal-2026-65E6/file/ICML%20Rebuttal%202026.pdf?v=31d0204e)):
>
> | Method   |   100 |   500 |   1000 |   5000 |   10000 |
> |:--------------|------:|------:|-------:|-------:|--------:|
> | KVP           | 10.67 | 16.32 |  19.06 |  26.44 |   28.75 |
> | JudgeQ        |  6.48 |  5.61 |   7.75 |   8.98 |   13.18 |
> | KeyDiff       |  2.85 |  4.29 |   7.92 |  15.14 |   12.75 |
> | SnapKV        |  8.26 | 11.86 |  12.78 |  14.47 |   16.09 |
>
>
> LongBench passage retrieval EN (See [Figures 6 and 7](https://anonymous.4open.science/api/repo/icml-rebuttal-2026-65E6/file/ICML%20Rebuttal%202026.pdf?v=31d0204e)):
>
> | Method   |   50 |   100 |   500 |   1000 |   5000 |   10000 |   20000 |
> |:--------------|-----:|------:|------:|-------:|-------:|--------:|--------:|
> | KVP           | 0.07 |  0.21 |  0.78 |   0.94 |   0.98 |    0.98 |    0.98 |
> | JudgeQ        | 0.03 |  0.05 |  0.33 |   0.62 |   0.98 |    0.96 |    0.97 |
> | KeyDiff       | 0.03 |  0.04 |  0.25 |   0.48 |   0.95 |    0.97 |    0.97 |
> | SnapKV        | 0.04 |  0.04 |  0.14 |   0.34 |   0.96 |    0.97 |    0.98 |
>
>
> At extreme compression (budget 500, ~4% retention), KVP achieves 0.78 vs. 0.27 for the next best baseline, demonstrating strong generalization well beyond its training distribution (limited to 4k context lengths).

---

> > ### Author Rebuttal · Reviewer_774D · 2026-03-31
> >
> > I thank the authors for their thorough rebuttal. I have read the response and appreciate the additional clarifications and comparisons provided. I maintain my evaluation.

---

### Official Review · Reviewer_NYhw · 2026-03-12

**Soundness:** 3
**Presentation:** 3
**Significance:** 3
**Originality:** 3
**Overall Recommendation:** 4
**Confidence:** 3

**Summary:**

The paper reframes KV cache eviction as a reinforcement learning (RL) problem: learning to rank tokens by their predicted usefulness for future decoding. To this end, this paper introduces KV Policy (KVP), a framework of lightweight per-head RL agents trained on pre-computed generation traces using only key and value vectors. Each agent learns a specialized eviction policy guided by a holistic reward, derived from future utility, that evaluates the quality of the ranking across all cache budgets, requiring no modifications to the underlying LLM or additional inference.

**Compliance With Llm Reviewing Policy:**

Affirmed.

**Final Justification:**

most of the my concerns are well addressed.

**Key Questions For Authors:**

Please refer to the section of "Strength and Weakness". Besides the questions listed there, a minor question includes:

1. When reporting the wall-clock time of KVP, does the reported time include the overhead of actually evicting tokens, such as data copying and tensor reconstruction?

**Limitations:**

Not enough.

**Strengths And Weaknesses:**

The introduction of query-free future utility is interesting, and the proposed method based on it appears reasonable. However, it is also the aspect that raises the most concerns for me.

1. The experimental evaluation is not sufficiently comprehensive. For a KV-cache compression or eviction paper in 2026, reporting the main results on only 4~5K context lengths, with very limited evaluation at 10K context (e.g., *GovReport* in *LongBench*), is far from convincing. A more complete evaluation on 128K or longer contexts and using the full set of subtasks in the RULER dataset would be necessary to demonstrate that the proposed learning objective and method generalize well across different query types and downstream tasks.

2. The authors should compare not only with the training-free methods, but also the learnable KV cache eviction methods, such as LOCRET (https://arxiv.org/pdf/2410.01805) and others.

---

> ### Author Rebuttal · Authors · 2026-03-31
>
> We thank the Reviewer for their review.
> We address the concerns with additional experiments on longer context lengths and learned KV cache eviction methods.
>
> ## 1. Evaluation at Longer Context Lengths
>
> We thank the reviewer for this suggestion. We now include evaluation at substantially longer contexts.
>
> **RULER at 128K context length.** We evaluate KVP on RULER at 128K context length. Not only KVP maintains strong performance at this scale, but even shows bigger improvements at extreme context lengths compared to the baselines:
>
> | Method   |   100 |   500 |   1000 |   5000 |   10000 |
> |:--------------|------:|------:|-------:|-------:|--------:|
> | KVP           | 10.67 | 16.32 |  19.06 |  26.44 |   28.75 |
> | JudgeQ        |  6.48 |  5.61 |   7.75 |   8.98 |   13.18 |
> | KeyDiff       |  2.85 |  4.29 |   7.92 |  15.14 |   12.75 |
> | SnapKV        |  8.26 | 11.86 |  12.78 |  14.47 |   16.09 |
>
> See [Figure 5](https://anonymous.4open.science/api/repo/icml-rebuttal-2026-65E6/file/ICML%20Rebuttal%202026.pdf?v=31d0204e).
>
> **LongBench passage retrieval EN.**  We include two new LongBench tasks with longer context, Passage Retrieval EN and ZH, and compute the retrieval score across cache budgets:
>
> | Method   |   50 |   100 |   500 |   1000 |   5000 |   10000 |   20000 |
> |:--------------|-----:|------:|------:|-------:|-------:|--------:|--------:|
> | KVP           | 0.07 |  0.21 |  0.78 |   0.94 |   0.98 |    0.98 |    0.98 |
> | JudgeQ        | 0.03 |  0.05 |  0.33 |   0.62 |   0.98 |    0.96 |    0.97 |
> | KeyDiff       | 0.03 |  0.04 |  0.25 |   0.48 |   0.95 |    0.97 |    0.97 |
> | SnapKV        | 0.04 |  0.04 |  0.14 |   0.34 |   0.96 |    0.97 |    0.98 |
>
> See [Figure 6 and 7](https://anonymous.4open.science/api/repo/icml-rebuttal-2026-65E6/file/ICML%20Rebuttal%202026.pdf?v=31d0204e).
>
> At extreme compression (budget 500, retaining \~4% of context), KVP achieves 0.78 vs. 0.27 for the next best baseline. These results demonstrate that KVP generalizes effectively to contexts far exceeding its training length (\~4.5K).
>
> We note that the original submission already evaluates beyond 4-5K: GovReport up to ~10K (Figure 4), zero-shot transfer at varying lengths (Figures 3, 4, 11), all 13 RULER subtask breakdowns (Figure 13), and cross-model generalization on Phi-4 14B (Figure 7).
>
> ## 2. Comparison with Learnable KV Cache Eviction Methods
>
> We thank the reviewer for suggesting a comparison with learnable methods. We address LOCRET and present a comparison with JudgeQ, a more directly comparable learned baseline.
>
> **LOCRET (Huang et al., 2025)** injects retaining heads into each attention layer that require [Q, K, V] as input. Since query vectors are not stored in the KV cache, this couples eviction to the LLM's forward pass. KVP operates solely on the KV cache (keys, values, positions), making it independent of the LLM forward pass and usable as a drop-in post-hoc module. A direct comparison would not isolate the eviction strategy from these architectural differences. We will discuss LOCRET in the revised related work as a complementary approach with different design tradeoffs.
>
> **JudgeQ (Liu et al., 2025)** provides a fairer comparison: both methods use the same ground-truth signal (generation attention), but JudgeQ learns from it via supervised MSE regression on attention maps with soft tokens inside the LLM, still requiring full forward passes; unlike KVP's fully offline RL approach on pre-collected traces. We reimplemented JudgeQ following the paper (no public code was available) and trained it on the same data using the same base model.
>
> On long-context evaluation, KVP substantially outperforms JudgeQ: on LongBench passage retrieval (Section 1), KVP achieves 0.78 vs. 0.25 at cache budget 500 and 0.93 vs. 0.70 at budget 1K. On RULER at 128K (Section 1), KVP likewise maintains a clear advantage.
> These results are confirmed on the short-context BoolQ benchmark ([Figure 8](https://anonymous.4open.science/api/repo/icml-rebuttal-2026-65E6/file/ICML%20Rebuttal%202026.pdf?v=31d0204e)).
>
> KVP consistently outperforms JudgeQ, and all other baselines, across all settings while being query-free and fully offline-trained.
>
> ## 3. Wall-Clock Time
>
> The timings in Appendix in Figure 6 measure the time to compute importance scores for a single KV head. They do not include physical tensor reconstruction, as this operation is implementation-specific and shared across all methods in the compress-once setting. We will clarify this.
>
> ## 4. Limitations
>
> The paper discusses the reliance on future attention as a proxy for task reward (Section 3.1.1), adaptive budget allocation and cross-head coordination as a future direction (Section 5). The compress-once evaluation setting (Section 4) is now partially addressed by our new long-context experiments, which use a chunked prefill-compress loop. We will consolidate these into a dedicated Limitations section.

---

> > ### Author Rebuttal · Reviewer_NYhw · 2026-04-03
> >
> > Most of my concerns are resolved, I will update my score accordingly.

---

### Official Review · Reviewer_X8J6 · 2026-03-15

**Soundness:** 3
**Presentation:** 3
**Significance:** 3
**Originality:** 3
**Overall Recommendation:** 4
**Confidence:** 3

**Summary:**

This paper studies KV-cache eviction for LLM inference and proposes to cast it as a learning-to-rank problem. The method, KVP, trains lightweight per-head RL agents that score tokens using only keys, values, and positions, and learns a single budget-agnostic ranking via a reward aggregated across all cache sizes. The empirical results on Qwen2.5-7B-Chat, with additional appendix experiments on Phi-4, are strong: KVP performs well on RULER and OASST2 and shows encouraging zero-shot transfer to downstream tasks such as BoolQ, ARC, and GovReport. Overall, a central concept addressed by the manuscript is whether KV eviction can be learned as a forward-looking ranking policy rather than engineered through heuristics.

**Compliance With Llm Reviewing Policy:**

Affirmed.

**Key Questions For Authors:**

Why is KVP evaluated only in the compress-once-after-prefill setting?
In long generations, newly generated tokens will also accumulate in the KV cache. Would KVP naturally extend to repeated compression during decoding, or is the method specifically intended only for prefill-time compression?

**Limitations:**

Yes

**Strengths And Weaknesses:**

Strengths

The reduction from subset selection to a single global ranking under uniqueness and nestedness is elegant, and the resulting budget-agnostic formulation is conceptually appealing. The paper gives a reasonable motivation for RL, and the use of a Plackett–Luce policy with Gumbel-sort yields an efficient way to train over permutations. The offline RL design is also practically attractive. At inference time, the policy uses only static token features (K/V/position), does not require query features or attention recomputation, and does not modify the base LLM. The method is evaluated on both synthetic long-context reasoning and multi-turn dialogue, and the reported curves across cache budgets are informative. The zero-shot transfer results are also a meaningful plus. This manuscript's general area comprises learned systems methods for efficient LLM inference, and within that area the paper is technically solid and empirically competitive.

Weaknesses

1. KVP is trained using future attention mass as token importance, aggregated into a per-budget eviction cost and then summed over budgets. This is a more forward-looking signal than common heuristics, but it is still not the same as directly optimizing task reward such as accuracy, perplexity, or generation quality.

2. The uniqueness and nestedness assumptions make the reduction elegant, but they are not empirically validated for real downstream task rewards. In practice, the paper defines a reward that makes ranking natural, rather than showing that true downstream utility is intrinsically consistent with a single ranking across all budgets. This is a reasonable modeling choice, but I do not think it is as “natural” as the paper sometimes suggests.

3. All online evaluations first build the full KV cache during prefill, then compress it once to the target budget, and only then start generation. It does not address settings where KV growth continues during decoding, where compression may need to be repeated, or where token importance drifts over multiple turns. This seems especially important because the method is presented as a KV-cache management policy, while the experiments more specifically evaluate pre-generation cache compression.

4. The paper emphasizes that KVP is efficient at inference because it avoids attention recomputation and uses only static token features. That is true for the online deployment setting they evaluate. However, the overall method also requires offline trace collection and RL training on stored Q/K/V traces. I do not think this invalidates the method, but the efficiency claims would be more balanced if the paper more clearly distinguished inference-time efficiency from full lifecycle cost.

---

> ### Author Rebuttal · Authors · 2026-03-31
>
> We thank Reviewer for their thoughtful and balanced review. We address each concern below.
>
> ## 1. Reward Based on Future Attention Mass vs. Task Reward
>
> We agree that future attention is a proxy, not a direct optimization of task reward. This is an intentional design choice: directly optimizing task reward would require running the full LLM within the RL training loop, contradicting our goal of lightweight, offline training.
>
> Crucially, future attention has a strong theoretical grounding: a KV entry that receives zero future attention is mathematically never used during generation, evicting it is provably lossless. More generally, future attention provides a forward-looking signal that is empirically well-correlated with downstream performance, as demonstrated by zero-shot transfer to BoolQ, ARC-Challenge, GovReport, MMLU, and HellaSwag (Figures 3, 4, 11) and much longer context lengths (up to 128k from the 4k seen during training).
>
> We further validate this with a comparison against JudgeQ (Liu et al., 2025; reimplemented; no public code available), a learned eviction method that trains soft tokens via supervised MSE regression on the same generation attention signal used by KVP, but requiring full LLM forward passes during training.
> Despite this, KVP outperforms JudgeQ across all settings.
> This is consistent with our supervised baselines analysis (see response to Reviewer tB21), where regression-based approaches on the same signal consistently underperform RL.
>
> ## 2. Compress-Once-After-Prefill Setting
>
> Our new long-context experiments directly address this concern. Both RULER at 128K and LongBench passage retrieval process context through a chunked prefill-compress loop, where KVP compresses the KV cache chunks. This implicitly evaluates KVP under repeated compression, not just a single prefill-compress-generate pass.
>
> RULER at 128K context length (see [Figure 5](https://anonymous.4open.science/api/repo/icml-rebuttal-2026-65E6/file/ICML%20Rebuttal%202026.pdf?v=31d0204e)):
>
> | Method   |   100 |   500 |   1000 |   5000 |   10000 |
> |:--------------|------:|------:|-------:|-------:|--------:|
> | KVP           | 10.67 | 16.32 |  19.06 |  26.44 |   28.75 |
> | JudgeQ        |  6.48 |  5.61 |   7.75 |   8.98 |   13.18 |
> | KeyDiff       |  2.85 |  4.29 |   7.92 |  15.14 |   12.75 |
> | SnapKV        |  8.26 | 11.86 |  12.78 |  14.47 |   16.09 |
>
>
> LongBench passage retrieval EN (see [Figures 6 and 7](https://anonymous.4open.science/api/repo/icml-rebuttal-2026-65E6/file/ICML%20Rebuttal%202026.pdf?v=31d0204e)):
>
> | Method   |   50 |   100 |   500 |   1000 |   5000 |   10000 |   20000 |
> |:--------------|-----:|------:|------:|-------:|-------:|--------:|--------:|
> | KVP           | 0.07 |  0.21 |  0.78 |   0.94 |   0.98 |    0.98 |    0.98 |
> | JudgeQ        | 0.03 |  0.05 |  0.33 |   0.62 |   0.98 |    0.96 |    0.97 |
> | KeyDiff       | 0.03 |  0.04 |  0.25 |   0.48 |   0.95 |    0.97 |    0.97 |
> | SnapKV        | 0.04 |  0.04 |  0.14 |   0.34 |   0.96 |    0.97 |    0.98 |
>
>
> KVP maintains strong performance, dramatically outperforming all baselines at aggressive budgets. Furthermore, KVP's scoring function is architecturally compatible with continuous compression during decoding: it relies only on static token features (keys, values, positions), so scores for previously cached tokens remain valid and only newly generated tokens require scoring.
>
> ## 3. Full Lifecycle Cost
>
> We appreciate the suggestion. As detailed in Appendix A.2-A.3:
>
> - **Trace collection**: A single forward pass over ~10,500 training samples, computationally equivalent to standard inference.
> - **Agent training**: 112 agents, each ~650K parameters, <30 minutes on 8 H100 GPUs. Checkpoint: 2.6MB per agent.
> - **Inference overhead**: ~1% prefill cost increase, zero overhead during autoregressive generation (Figure 6).
>
> This one-time cost is amortized over the deployed model's lifetime.

---

> > ### Author Rebuttal · Reviewer_X8J6 · 2026-04-06
> >
> > The rebuttal has adequately addressed my concerns, and I will maintain my positive score.

---

### Decision · Program_Chairs · 2026-04-30

**Decision:**

Accept (regular)

**Comment:**

This paper reframes the problem of Large Language Model (LLM) KV-cache eviction from a binary keep/evict heuristic into a learning-to-rank problem. The authors introduce KVP (Key-Value Policy), which trains lightweight, per-head reinforcement learning agents to rank tokens based on their predicted future utility. By relying exclusively on static cache features (keys, values, and positional data) and eliminating the need for query-dependent attention recomputation, the method offers a highly efficient inference-time footprint.

The reviewers commend the paper for its elegant conceptual shift, its budget-agnostic formulation, and its strong empirical results—including encouraging zero-shot transfer capabilities on downstream tasks. While the reviewers are recommending acceptance based on these merits, reviewers raised critical concerns regarding the mathematical formalization of the reward, the theoretical claims of NP-hardness, and the scale of the evaluation. We highly recommend authors address these points in the revision.